# Oxygen Deficient (OD) Combustion and Metabolism: Allometric Laws of Organs and Kleiber's Law from OD Metabolism?

**Kalyan Annamalai**

Department of Mechanical Engineering, Texas A&M University, College Station, TX 77843-3123, USA; k-annamalai@tamu.edu

**Abstract:** The biology literature presents allometric relations for the specific metabolic rate ($SMR_k$) of an organ k of mass mk within the body of mass $m_B$: $SMR_k \propto m_B^{fk}$ (body mass allometry, BMA). Wang et al. used BMA, summed-up energy from all organs and validated Kleiber's law of the whole body: $SMR_M \propto m_B^{b'}$, b' = −0.25. The issues raised in biology are: (i) why $f_k$ and b' < 0, (ii) how do the organs adjust $f_k$ to yield b'? The current paper presents a "system" approach involving the field of oxygen deficient combustion (ODC) of a cloud of carbon particles and oxygen deficient metabolism (ODM), and provides partial answers by treating each vital organ as a cell cloud. The methodology yields the following: (i) a dimensionless "group" number $G_{OD}$ to indicate extent of ODM, (ii) $SMR_k$ of an organ in terms of the effectiveness factor; (iii) curve fitting of the effectiveness factor to yield the allometric exponents for the organ mass-based allometric laws (OMA); (iv) validation of the results with data from 111 biological species (BS) with $m_B$ ranging from 0.0075 to 6500 kg. The "hypoxic" condition at organ level, particularly for COVID-19 patients, and the onset of cancer and virus multiplication are interpreted in terms of ODM and glycolysis.

**Keywords:** oxygen deficiency (OD); metabolism; effectiveness factor; vital organs; allometric laws; COVID-19; cancer

## 1. Introduction and Background

Air breathing thermal power systems, such as automobile engines, convert chemical energy into thermal energy via combustion of fuels (a high temperature oxidation) of fuels with $O_2$ supplied from the air. Figure 1a illustrates a multi-cylinder internal combustion engine supplied with premixed fuel and air. A part of the thermal energy in the engine is converted to useful mechanical work $\{\dot{W}\}$. The remaining energy is disposed of as heat loss $\{\dot{Q}\}$. The ratio of work output to energy input is termed as the thermal efficiency, which is about 35%. Air-breathing biological species (BS), including humans, are in many ways like those thermal systems and calorimetry and thermodynamics are strongly coupled to metabolic processes within BS [1]. Just as crude oil is refined into gasoline, diesel, and kerosene for use in thermal systems, the intestines of the digestion system of BS serve as a "food refinery" and convert food into three basic nutrients: Carbohydrates (CH, e.g., glucose), Fats (F, e.g., palmitic acid) and Proteins (P). The energy release rate (ERR, or $\dot{q}$, called metabolic rate, MR, in biology) occurs mainly through the oxidation of CH and F. Unlike rapid oxidation (combustion) of fuel in thermal systems, the BS uses a slow oxidation process to convert 2/3 of *chemical energy* $\{\dot{q}\}$ into heat $\{\dot{Q}\}$, and the remaining part is converted into "chemical work" $\{\dot{W}_{ATP}\}$ in the form of the production of adenosine triphosphate (ATP) within the organs of BS (Figure 1b). In the BS, the heat is used to overcome heat loss from BS to the surroundings and to maintain body temperature independent of the external environment (homeostasis). The body temperature of BS is

about 36 °C for elephants, 37 °C for humans, and 42 °C for chickens [2]. The ATP is known as the life-sustaining work currency of the body. It is used to (i) transport nutrients, (ii) maintain the thermodynamic potential for driving reactions and species transfer, (iii) supply energy for endergonic reactions required the functioning of various organs (e.g., work for pumping blood out of heart) and (iv) provide power to propel the BS. Concentration of ATP is used to detect the presence of virions [3]. The ratio $\dot{W}_{ATP}$ to $\dot{q}$ is called the metabolic efficiency ($\eta_M$), which is typically 35%, and is somewhat similar to the thermal efficiency of an automobile. Dobson's review on the scaling relation for BS and the estimate of lifetime ATP production seems to indicate an average metabolic efficiency of 32% [4].

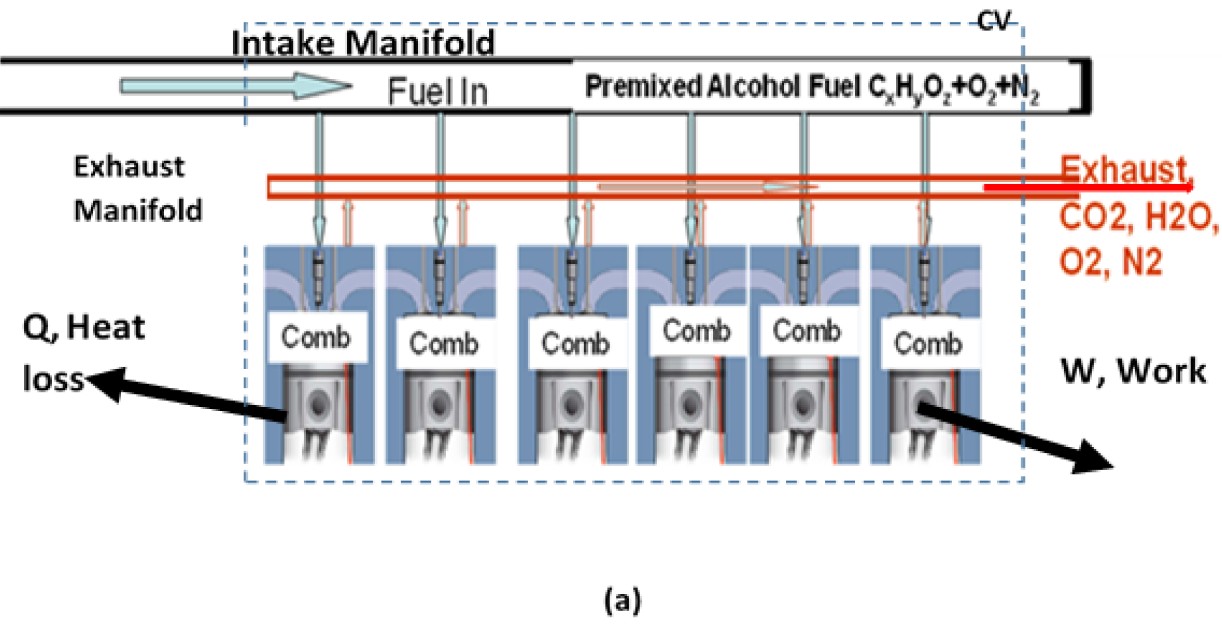

**(a)**

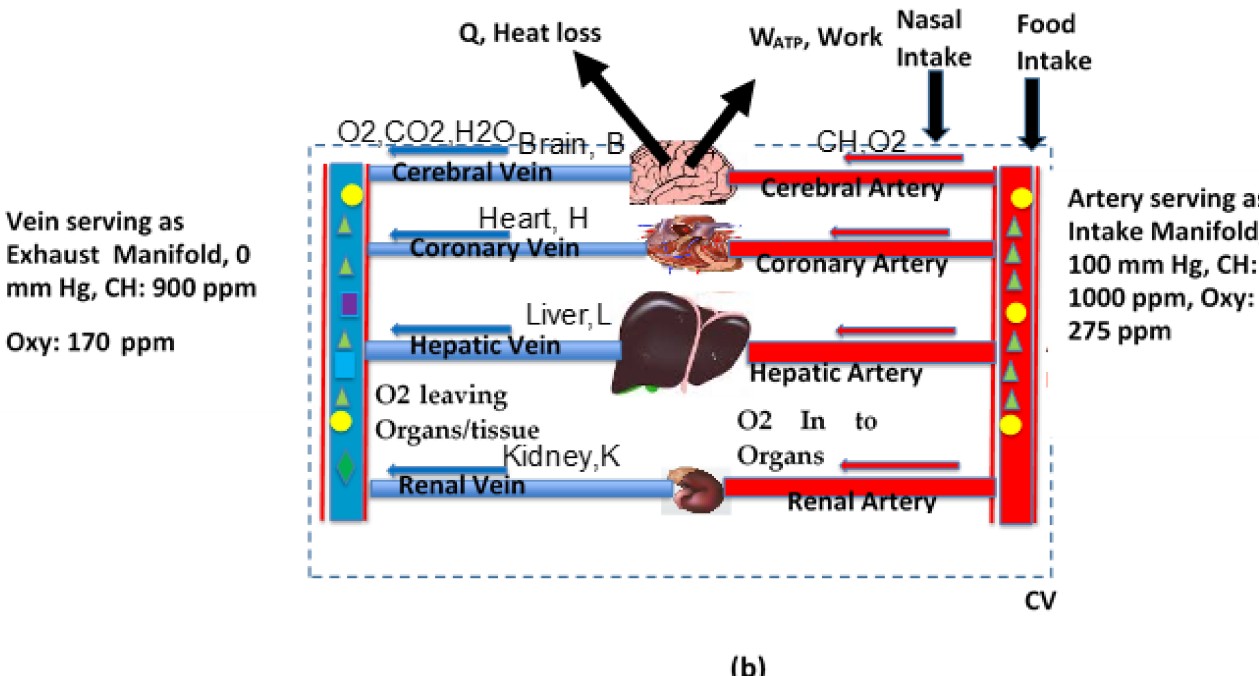

**(b)**

**Figure 1.** Illustration of energy conversion processes within Thermal Systems and Biological Systems. (**a**) Multi-Cylinder Automobile Engine: For the control volume (CV) indicated, fuel and $O_2$ enter the system, and exhaust products $CO_2$, $H_2O$

and unused $O_2$ leaves the system. A part of the energy released is converted into work (W) and the remainder is released as heat (Q); (**b**) Multi-organ BS: Each organ is like a cylinder in an automobile engine. For the control volume (CV) indicated, food enters through mouth and is converted into nutrients CH, F, and P, and $O_2$ breathed through the nose joins the blood stream. Thus, premixed fuels CH, F, and $O_2$ enter the blood stream and are transported to vital organs Br, H, K, and L, Exhaust products, $CO_2$, $H_2O$, and unused $O_2$ leaves the blood stream, enters the lungs, and become exhausted. A part of the energy released from each organ is converted into ATP (equivalent to work, W) and the remainder is released as heat (Q). $p_{A,O2}$ in Alveoli: 104, $p_{O2}$ in tissue capillaries is typically 40–50 mm Hg with $O_2$ {saturation level almost at 60%}.

The nutrients CH, F, and P obtained from digestion of food intake pass through the liver to regulate the level of CH at about 1000 ppm (5.88 mM, assuming $\rho_{bl}$ = 1.060 kg/L) in the blood stream of inferior vena cava and the excess CH is stored as glycogen. Unlike CH, the $O_2$ cannot be stored except as oxyhemoglobin (OHb) in 5 L of human blood and hence there is a constant need for oxygen through respiration. The $O_2$-poor blood stream from inferior and superior vena cava enters the heart and is pumped to the lungs for enrichment with $O_2$, which is obtained through air breathed into the air sacs (alveoli, a vascularized sac) of the lungs. The volume of air drawn in per breath, the tidal volume, like the air drawn in during one stroke of an automobile engine, is typically 500 mL for 70 kg per person or about seven mL per kg of body mass. The $O_2$ exists at partial pressure of $p_{A,O2}$ in the alveoli. The $O_2$ is transferred from the alveoli to blood capillaries, and is subsequently dissolved into the blood where it exists at partial pressure of $p_{a,O2}$. From thermodynamics [5], the relation for concentration of dissolved $O_2$ ($O_2$, (aq)) is given by Henry's law:

$$[O_2 \text{ (aq)}] \left\{ \frac{\text{mL of } O_2}{\text{L of Blood}} \right\} = H_{O2} \left\{ \frac{\text{mL of } O_2}{\text{L of Blood mm Hg}} \right\} p_{O2} \text{ (mm Hg)}, \quad p_{O2} = p_{A,O2} \approx p_{a,O2} \quad (1)$$

where $H_{O2}$ is Henry's constant. The $H_{O2}$ is 0.021 to 0.037 mL of $O_2$ per L of blood per mm of Hg [6,7]. See Ref [8] for the approximation $p_{A,O2} \approx p_{a,O2}$. The $p_{A,O2}$ is about 105 mm Hg while $p_{a,O2}$ ranged from 95–100 mm Hg. While the biology literature presents oxygen content as mL of gas (CST) per L of blood, combustion literature presents this as mass fraction ($Y_{O2}$). Hence Equation (1) for dissolved oxygen is converted into the oxygen mass fraction basis:

$$Y_{O2} \left\{ \frac{\text{g of } O_2}{\text{g blood}} \right\} = H_{O2} \left\{ \frac{\rho_{O2}}{\rho_{bl}} \right\} * 10^{(-3)} \, p_{O2} \text{ (mm Hg)} = 4.15 \times 10^{(-8)} \left\{ \frac{\text{g of } O_2}{\text{g blood mm Hg}} \right\} p_{O2} \text{ (mm Hg)} \quad (2)$$

Selecting $H_{O2}$ = 0.031 mL of $O_2$ per L per mm Hg.

Once in blood stream, the dissolved $O_2$ binds with the hemoglobin (Hb) in the red blood cells (RBC, 350 million Hb molecules per RBC) to produce oxy-hemoglobin (OHb) or $Hb(O_2)_n$, n = 1, 2, 3, 4. Thus, about 1.4 billion oxygen molecules are packed per RBC if n = 4. Eventually, chemical equilibrium is reached for Hb oxidation reactions within venous blood leaving the lungs. The equilibrium reactions are given by:

$$Hb + n \, O_2 = Hb \, (O_2)_n, \quad n = 1, 2, 3, 4 \quad (3)$$

$$K_n'^0 \left\{ \frac{1}{(\text{mm Hg})^n} \right\} = \left\{ \frac{[Hb \, (O_2)_n]}{[Hb] \, p_{O2}^n} \right\}, \quad p_{O2} \text{ in mm Hg}, \quad n = 1, 2, 3, 4 \quad (4)$$

The equilibrium constants $K_n^0$ are given as [9]

$$K_1'^0 = 0.01 \, \{\text{mm Hg}\}^{-1}, \, K_2'^0 = 0.02 \, \{\text{mm Hg}\}^{-2}, \, K_3'^0 = 0.04 \, \{\text{mm Hg}\}^{-3}, \, K_4'^0 = 0.08 \, \{\text{mm Hg}\}^{-4}$$

The square parentheses in Equation (4) represent molal concentrations. The ratio of oxygen captured as OHb to the maximum that could be captured if all Hb in the blood is oxidized to $Hb(O_2)_4$ is the saturation fraction. The equilibrium calculations yield the plot of saturation percentage and concentration of OHb (n = 1, 2, 3, and 4) vs. $p_{O2}$ (Figure 2).

The blood, which now contains the premixed CH: $O_2$ mixture, is pumped by the heart through large vessels (called macro vasculature) to various organs within the body.

With $\rho_{bl}$ = 1060 g/L, $H_{O2}$ = 0.031 mL of $O_2$ per L of blood and per mm of Hg, $\rho_{O2}$ = 1.42 g/L, and 1 g of Hb holds 1.36 mL of $O_2$ in $Hb(O_2)_4$, the total oxygen {dissolved and oxygen bound oxyhemoglobin) content in blood can be shown to be:

$$\text{Total oxy}, Y_{O2}, \text{ ppm} = 0.182 \text{ (Hb)}\left(\frac{g}{DL}\right) Sa_{O2}\% + 0.0415 \text{ } p_{a,O2} \text{ (mm Hg)}. \qquad (5)$$

where $Sa_{O2}$ is saturation % and (Hb) is mass concentration. See Ref. [10] also for oxygen content in terms of mL of $O_2$ per L of blood. Equation (5) indicates that the oxygen content in blood is controlled by three parameters: (Hb), Saturation % and arterial $p_{a,O2}$. Under normal conditions with arterial blood saturation, $SaO_2$ = 95%, $p_{aO2}$ = 100 mm of Hg, (Hb) = 15 g/dL {typically, male: 14–17 g/dL, female:12–15), dissolved oxygen is computed as 4 ppm, oxygen bound to Hb is 260 ppm {oxyhemoglobin, OHb or $Hb(O_2)_n$, n = 1, …, 4}; i.e., total oxygen content of 264 ppm is dominated by oxy-Hb and hence almost proportional to Hb content of blood at given $p_{O2}$ and $SaO_2$%. The OHb serves as the carrier of oxygen. Figure 2 uses equilibrium constants and hence the total oxygen content at equilibrium is slightly different from Ref. [10] and is about 275 ppm {4.5 ppm as dissolved $O_2$, 270 ppm as oxyhemoglobin (OHb or $Hb(O_2)_n$}.

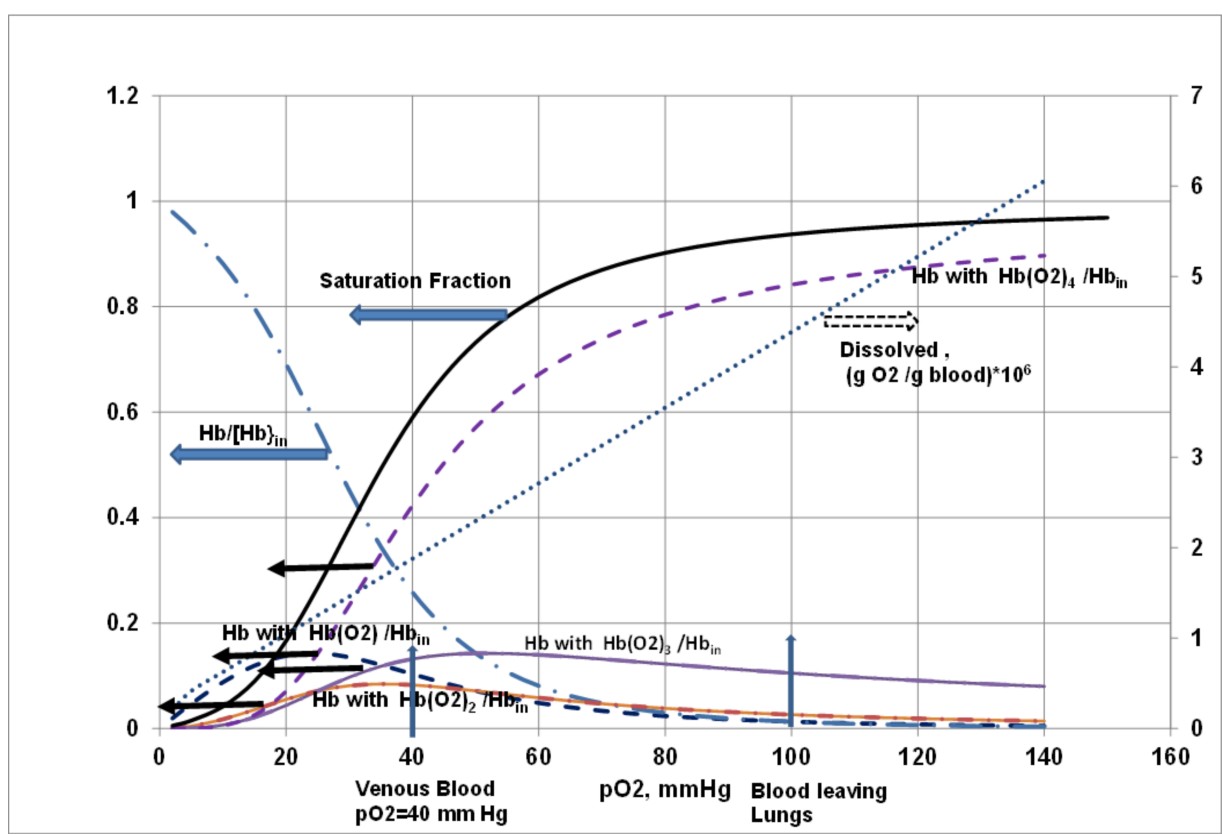

**Figure 2.** Variation of the (i) saturation fraction, (ii) $(Hb(O_2)_n)/(Hb)_{in}$, (iii) dissolved $O_2$, and (iv) amount left over (Hb)/(Hb) in $p_{O2, A}$ in the alveoli ($\approx p_{O2,a}$ in the artery of the blood stream). Dissolved $Y_{O2}$ in the blood (g per g blood = $4.2 \times 10^{-8}$. The saturation fraction is 0.94 at $p_{O2,A} \approx p_{O2,a}$ = $p_{O2}$ = 100 mm of Hg. Normal saturation levels: 92–98%; hypoxic if Sat.% < 90%; organ damage if Sat. < 80%, loss of consciousness if Sat. 75%, In COVID-19 patients, saturation level ranges from 50–90% [11] due to "excessive" clotting in blood vessels. Figure 2 adopted from [12] and modified. Normal $(Hb)_{in}$ = 160 mg per L of blood and 3 mL of dissolved blood per L of blood (1 mL of $O_2$ at CST = 1.42 mg).

Multi-Cylinder Automobile vs. Multi-organ BS: The fuel used in a multi-cylinder automobile requires stoichiometric oxygen: fuel ($v_{O2,st}$) mass ratio of 3.5 for complete com-

bustion. However, a lean premixed mixture {typical oxygen: fuel (octane) ratio: 4.2} is supplied to each cylinder for combustion and the unused oxygen, along with products of complete combustion ($CO_2$ and $H_2O$), are exhausted (Figure 1).

(a). On the other hand, the multi-organ BS is supplied with arterial blood containing a rich premixed mixture {typical oxy: fuel (CH or glucose) ratio: 0.3 while stoichiometric oxygen: fuel ratio ($\nu_{O2,st}$) is close to unity (Table 1, Section 5.1)}.

**Table 1.** Stoichiometric Oxygen and Air, HV, $HV_{O2}$, $HV_{O2'}$ of Nutrients and Fuels.

| Fuel | $O_2$/Fuel Stoich. Ratio, $\nu_{O2, st}$ | | Air/fuel Mass Ratio | RQ [a] | HV (Per Fuel) [b] | | $HV_{O2}$ (Per $O_2$) | | |
|---|---|---|---|---|---|---|---|---|---|
| | Molar | Mass | | | (kJ/mol) | (kJ/g) | (kJ/mol) | (kJ/g) | (kJ/L) [c] |
| gasoline ($CH_{2.46}$) | 1.62 | 3.58 | 7.71 | 0.6 | 698.5 | 48.3 | 431.2 | 13.5 | 17.6 |
| methane ($CH_4$) | 2 | 4 | 9.52 | 0.5 | 890.3 | 55.5 | 445.2 | 13.9 | 18.2 |
| coal-dry, ash-free [d] ($CH_{0.714}O_{0.18}$) | 1.09 | 2.23 | 6.22 | 0.9 | 465.2 | 29.8 | 427.4 | 13.4 | 17.4 |
| cattle manure-dry, ash free [d] ($CH_{1.78}O_{0.64}N_{0.08}S_{0.014}$) | 1.1 | 1.42 | 5.43 | 0.9 | 530.2 | 20.0 | 431.0 | 14.0 | 18.3 |
| glucose [e] ($C_6H_{12}O_6$) | 6 | 1.07 | 28.56 | 1.0 | 2813 | 15.6 | 468.9 | 14.7 | 19.1 |
| Fat ($C_{16}H_{32}O_2$) | 23 | 2.87 | 12.39 | 0.7 | 10,015 | 39.13 | 436.3 | 13.6 | 17.8 |
| protein [d] ($C_{4.57}H_{9.03}N_{1.27}O_{2.25}S_{0.046}$) | 5.7 | 1.54 | 9.61 | 0.8 | 2718 | 22.8 | 476.7 | 14.9 | 19.5 |

[a] RQ, Respiratory quotient = $CO_2$ moles/$O_2$ moles. [b] All heating values are higher heating values. For fuels with empirical formula, values per empirical mol. $\Delta H_R^\circ = -HV$. [c] Based on SATP volume. For conversion from kJ/g to kJ/SATP multiply L by 1.30, and to kJ/CSA, multiply L by 1.42. [d] Empirical Chemical Formulae. [e] 1 g of $O_2$ consumption releases 13.6 to 14.7 kJ of energy for F and CH (the average is about 14.2 kJ per g or 20.2 kJ/CSA L of $O_2$ consumed).

The cells of each organ are irrigated with CH, $O_2$ by arterial blood and products of oxidation, then $CO_2$ and $H_2O$ are released to the venous blood (saturation $O_2$ percentage leaving the organs: 65–75%). Thus, the vein of each organ serves as an exhaust manifold containing unused oxygen and CH along with products of oxidation (Figure 1).

(b). The heart pumps the $O_2$ poor blood to the lungs for breathing out of $CO_2$ and $H_2O$ along with unused oxygen and thus replenishing the blood stream with the required $O_2$.

Body Mass Based Allometry {BMA} for Energy Release Rate {SERR} of Organs and Whole Body: The oxygen extraction fraction (OEF, called the equivalence ratio (ER) in the combustion literature for lean mixtures) is defined as

$$\text{OEF} = \left\{\frac{O_2 \text{ used}}{O_2 \text{ inspired}}\right\} = \left\{\frac{O_2 \text{ inspired} - O_2 \text{ unused}}{O_2 \text{ inspired}}\right\} = \left\{\frac{O_2 \text{ used}}{O_2 \text{ used} + O_2 \text{ unused}}\right\} \quad (6)$$

Just as intake and exhaust gas analyses are used to determine the amount of $O_2$ consumed [13,14] in automobile engines, nasal intake and exhaust analyses are used to determine the OEF of BS. Typically, OEF for whole body of the order of 0.25. The ERR of the whole body in Watts is determined by multiplying the $O_2$ consumption rate of the body (g of $O_2$/s) by the heating value expressed in J per g of $O_2$ consumed ($HV_{O2}$). The $HVO_2$ is almost constant for most fuels and nutrients as shown in Table 1 [15].

Similarly, the $O_2$ consumption rate of an organ $k$ of mass $m_k$ (e.g., liver) is estimated by measuring the blood flow rate along with the oxygen concentration at the inlet and exit of an organ (or saturation %), estimating the $O_2$ consumption rate and then computing the $\underline{ERR}_k$ ($\dot{q}_k$) of each organ k. The specific metabolic rate ($SMR_k$) of an organ k ($\dot{q}_{k,m}$) is estimated using $\dot{q}_{k,m} = \dot{q}_k/m_k$ for many species with wide ranging body masses {$m_B$}. The

data for $m_k$, $\dot{q}_{k,m}$ vs. $m_B$, are then curve fitted in terms of body mass ($m_B$), called allometric laws in biology. The vital organs brain (Br), heart (H), kidney (K), and liver (L) perform life-sustaining functions [16]. The body mass based allometry (BMA} for $\dot{q}_{k,m}$ and $m_k$ for vital organs are given below in terms of the empirical allometric constants $c_k$, $d_k$, $e_k$, and $f_k$ (Table 2, ref. [17]):

$$\mathrm{SMR}_k = \dot{q}_{k,m} \left\{ \frac{\mathrm{Watts}}{\mathrm{kg\,of\,organ\,k}} \right\} = e_k\, m_B{}^{f_k},\ f_k < 0,\ k = K, H, Br, L, R \qquad (7)$$

$$m_k = c_k\, m_B{}^{d_k},\ d_k > 0,\ k = K, H, Br, L, R \qquad (8)$$

and $f_k = 0$ for isometric law. The coefficients $c_k$ and $e_k$ are the allometric pre-exponents and $d_k$ and $f_k$ are the allometric exponents. The studies by Wang et al. provide a heterogeneous or reductionist approach where the whole-body metabolic rate is estimated by summing up contributions from all organs. Wang et al. [17] used Equations (7) and (8) and body mass based allometric constants (Table 2) to estimate the metabolic rate of each organ and then computed the metabolic rate of the whole body as a sum of the metabolic rates of all organs, $\dot{q}_{Het}$.

$$\mathrm{ERR\ or\ MR,}\ \dot{q} = \dot{q}_{Het} = \sum_k \dot{q}_{k,m}\, m_k = \left( \sum_k c_k\, e_k\, m_B{}^{d_k+f_k} \right)$$
$$m_B = \sum_k m_k,\ k = Br, H, K, L, R \qquad (9)$$

When $\dot{q}_k$ is computed by summing the metabolic rates within the whole body, it is denoted as $\dot{q}_{Het}$ (see Table A1 in Appendix B). The variation of MR of the whole body ($\dot{q}$) with body mass is curve fitted using the following allometric relation, ref. [17]:

$$\dot{q} = a\, m_B{}^b,\ a = 3.25 \left( \frac{W}{\mathrm{kg}^{0.76}} \right),\ b = 0.76 \qquad (10)$$

The relation given by Equation (10) is known as Kleiber's law in biology. The above constants a and b are close to the constants determined by Kleiber, who considered a variation of the whole-body metabolic rates of animals (homogeneous or holistic approach) with body size (mass) changing by a factor of 2800 and found that $a = 3.55\ \mathrm{W/kg}^{0.74}$ and $b = 0.74$ [18]. Dividing $\dot{q}$ by body mass, $m_B$, the specific energy release rate (W/kg body mass), $\mathrm{SERR}_M$ (or $\mathrm{SMR}_M$, known as specific metabolic rate in biology) is given as,

$$\mathrm{SERR}_M \text{ or } \mathrm{SMR}_M,\ \dot{q}_M \left( \frac{\mathrm{Watts}}{\mathrm{kg\,body\,mass}} \right) = \left( \frac{\dot{q}}{m_B} \right) = a\, m_B{}^{b-1} = a\, m_B{}^{b'} \qquad (11)$$

where the subscript M denotes SMR based on the whole body $m_B$. Kleiber's law indicates a higher $\mathrm{SMR}_M$ for smaller species (e.g., ant, baby, rat, etc.).

From Table 2, it is observed that the liver (L) is the largest vital organ, and the kidney (K) is the smallest organ in the body. The kidney, with 0.5% body mass, consumes 10% of the $O_2$ consumed by the whole body. The ERR from all of the vital organs (BrHKL) of a 70 kg person having a combined mass of 3–4 kg (about 5% of total body mass) is estimated at 30 W (40% ERR from the whole body) while the total ERR, or the basal rate (BMR) is about 82 W. The residual mass R (i.e., 96% mass) contributes to the remaining 60%.

OEF of Organ k: The oxygen used by organ k is given as

$$O_2 \text{ used by organ k} = \dot{m}_{O2,k}\ (\mathrm{kg/s}) = \left( \frac{\dot{q}_{k,m} m_k}{HV_{O2}} \right) = \left( \frac{c_k e_k}{HV_{O2}} \right) m_B{}^{d_k+f_k} \qquad (12)$$

Following Equation (7), the oxygen extraction fraction (OEF) or equivalence ratio (ER) of organ k is given as:

$$(OEF)_k \text{ or } (ER)_k = \left\{ \frac{O_2 \text{ used}}{O_2 \text{used} + O_2 \text{ unused}} \right\}_k = J_k \; m_B^{L_k} \tag{13}$$

**Table 2.** Allometric Constants for Organ Mass, Energy Release (Metabolic) Rate. Values based on 6 species ranging in mass from 0.45 kg to 65 kg. he body is composed of four vital organs Br, H, K, L with the fifth organ bei.ng the rest of the body (R); $c_k, d_k, e_k$, and $f_k$ are from [17]; density from [19]). $F_k$ values from Ref. [20].

| Organ | Density, $\rho_k$, g/cc | $c_k$, kg | $d_k$ | $e_k$ f | $f_k$ g | $m_k$ in kg for 85 kg Human | $E_k$ | $F_k$ | $q_{k,m}$ for 85 kg Human | $J_k$ (Assumed $Y_{O2in}$ = 300 ppm) [20] | $L_k$ [20] | $OEF_k$ for 84 kg Human |
|---|---|---|---|---|---|---|---|---|---|---|---|---|
| Brain (Br) | 1.036 | 0.011 | 0.76 | 21.62 | 0.14 | 0.32 | 9.42 | −0.184 | 0.044 | 0.524 | −0.084 | 0.37 |
| Heart (H) | 1.06 | 0.006 | 0.98 | 43.11 | 0.12 | 0.47 | 23.04 | −0.122 | 0.15 | 0.257 | −0.146 | 0.48 |
| Kidneys (K)h | 1.05 | 0.007 | 0.85 | 33.41 | 0.08 | 0.31 | 20.94 | −0.094 | 0.11 | 0.0863 | −0.004 | 0.085 |
| Liver (L) | 1.06 | 0.0330 | 0.87 | 33.11 | 0.27 | 1.57 | 11.49 | −0.310 | 0.19 | 1.558 | −0.256 | 0.52 |
| Residual Mass (R)- | | 0.939 | 1.01 | 1.45 | 0.17 | 83.44 | 1.44 | −0.168 | 0.19 | - | - | |

$\dot{q}_{k,m'} \left\{ \frac{Watts}{kg \text{ of organ } k} \right\} = e_k \; m_B^{f_k} = E_k \; m_k^{F_k}, \; m_k = c_k \; m_B^{d_k}, F_k = \left\{ \frac{f_k}{d_k} \right\}, \; E_k = \left\{ \frac{e_k}{c_k^{\left( \frac{f_k}{d_k} \right)}} \right\}, \; k = K, H, Br, L, R; \; OEF_k = j_k \; m_B^{L_k}.$

f Elia values for "$e_k$" are: BrHLK and R: 11.62, 21.3, 9.7, 21.3, and 0.58 W/kg $f_k = 0$ [21]; Krebs report that SMR of organs decreases with an increase in body mass, and the order of decrease is the same as the decrease in SMR of the body [22]. g Ref. [23] cites $f_L = −0.17$ to 0.21 for Liver or Hepatocytes; kidney cortex: −0.11 to −0.07, brain: −0.07. h Gutierrez: kidneys $m_K \propto mB_{0.85}$; for liver $m_L \propto mB^{0.87 \text{ to } 0.89}$ [24].

The estimated allometric constants for $J_k$ and $L_k$ (based on data for the six species) along with $(OEF)_k$ are presented in Table 2. (See [20] for details).

The foregoing results from biology suggest that if all $f_k$ values are negative, this leads to an increased $SMR_k$ of each organ, or increased activity of vital organs for smaller species, with a corresponding increase in $SMR_M$ for smaller species which have a larger surface area to volume ratio (S/V). The higher S/V requires a higher $SMR_M$ to maintain the same body temperature. Biologists raise the following questions:

1. Specifically, when body mass increases across mammals, why are all $f_k$ values negative {Equation (7)}? Why do various organs and tissues have different $d_k$ and $f_k$ values? [17].

2. The allometric size relationship is somehow 'programmed' into cells although the factors that let them know whether they are in a small or large organism are still unknown [25].

In other words, how do the organs within the body know that they are in smaller or larger species and adjust $\dot{q}_M$ to yield $b'$ as a negative value {Equation (11)} and how do they set $b' = −0.26$ [17,25]? In order to explain these puzzles, a detailed literature review linking the fields of OD combustion of a group of carbon particles with OD metabolism in organs (a group of cells) is undertaken.

## 2. Literature Review of Group or Oxygen Deficient Combustion (ODC) in Engineering and Oxygen Deficient Metabolism (ODM) in Biology

Prior to the review of OD combustion of a cloud of carbon particles, an elementary overview of combustion of carbon particle is presented below.

### 2.1. Oxidation of Carbon

In the presence of $O_2$, the oxidation of carbon to $CO_2$ occurs in two steps: heterogeneous oxidation of C to CO {$\Delta H = −110.5$ kJ/mol CO, $\Delta G^0 = −137.2$ kJ/mol CO or C, and homogenous oxidation of CO to $CO_2$ {$\Delta H = −283.0$ kJ/mol $CO_2$ or CO, $\Delta G^0 = −257.2$ kJ/mol $CO_2$ or CO} where $\Delta H$ denotes the change in enthalpy of reaction, and $\Delta G^0$ denotes change

in the Gibbs function or the work potential energy. Thus % energy released by step-1is about 28%.

### 2.2. Group Combustion Or ODC

A single carbon particle of given particle diameter, $d_p$, burns quickly and consumes oxygen rapidly due to an abundance of oxygen. If one adds more and more particles (e.g., dust cloud), the oxygen consumption rate of the whole cloud increases due to a greater number of carbon particles. However, the increase is less than proportional to the increase in cloud mass or number of particles since the particles, particularly those near the core of the cloud, are at a lower concentration of $O_2$. Thus, the oxygen deficiency causes the $O_2$ consumption rate per unit mass of cloud (g of $O_2$ per s per g of cloud mass) to decrease.

The engineering literature extensively modeled the burn, and hence energy release, rate as a function of a cloud size, and particle size (Chapter 16 [26]). Consider the spherical dust cloud of radius R and number deity of "n" (particles/cm3) shown in Figure 3 with the cloud surface exposed to oxygen massfraction of $Y_{O2, cl}$. Figure 3 shows three regimes: (a) a diluted cloud where the $Y_{O2}(r)$ profile is almost flat with all particles exposed to $Y_{O2cl}$; (b) $Y_{O2}(r)$ decreases with a decrease in r when core particles are subjected to a low $Y_{O2}$; (c) a very dense cloud with outer particles in an aerobic shell and inner core particles receiving almost no oxygen (anerobic core). The combustion literature presents the conservation equations, boundary conditions, solutions for $O_2$ profiles and specific consumption rate of oxygen or SERR. They are summarized in column 3 of Table 3. The specific oxygen consumption, and hence SERR of the cloud (watts per g of cloud), decreases with an increase in the total mass of the cloud, $m_{Cl}$ [26] (i.e., when there are a larger number of particles within the cloud).

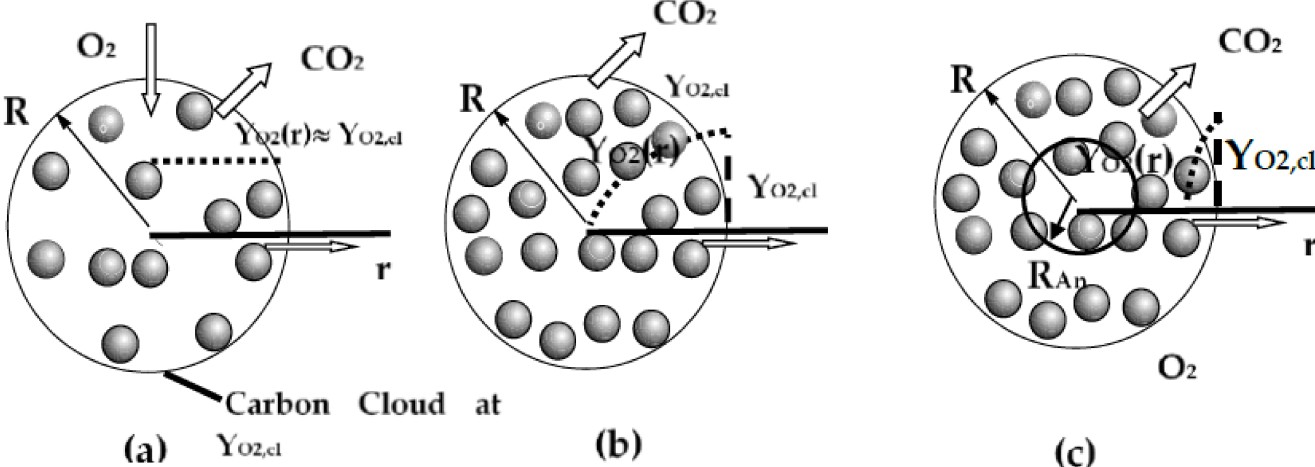

**Figure 3.** An example of the Spherical Carbon Cloud of radius R subjected to $Y_{O2, cl}$ or $Y_{O2, R}$ and cloud temperature, $T_{cl}$, at cloud surface; instead of $Y_{O2R}$, sometimes the dust cloud is exposed to an ambience at $Y_{O2\infty}$. (**a**) Diluted Cloud (*n* is extremely low): Isolated Combustion Mode with uniform $O_2$ concentration at $Y_{O2, cl}$. (**b**) Dense Cloud: Interactive Combustion Mode with a decreasing $O_2$ concentration within the cloud with a non-uniform $O_2$ consumption per unit volume. (**c**) Very Dense Cloud: Combustion of cloud with an anaerobic core of radius $R_{an}$ where the $O_2$ concentration is almost zero. Volume/surface area = R/3.

Chiu et al. formulated the group combustion (GC) model involving liquid drop clouds [27] with *number density (n, drops per cm³)* and introduced a non-dimensional G# to characterize the denseness of the cloud.

$$G = \frac{\text{Charactristic } O_2 \text{ consumption rate by all drops within drop cloud}}{\text{Charactristic } O_2 \text{ diffusion rate to the drops from cloud surface}}, \tag{14}$$

The G is the group combustion (GC) number. This is known as the Chiu number for drop clouds [28]. Annamalai et al. extended the model to carbon dust clouds for three geometries slab, cylinder, and spherical clouds [26] and generalized *G#* for carbon dust clouds releasing energy under kinetics or diffusion-controlled models:

$$G = \left\{ \frac{C_{ch,p} \, n \, R^2}{(\rho D)_{eff}} \right\} = \frac{\text{Charactristic } O_2 \text{ consumption rate by all particles within carbon cloud}}{\text{Charactristic } O_2 \text{ diffusion rate to the particles from cloud surface}}, \quad (15)$$

assuming that the $O_2$ consumption rate per carbon particle located at "r" is given as where the equation for $\dot{w}_{O2,p}$ is defined in row 1, column 3 of Table 3 and the characteristic constant $C_{Ch,p}$ changes depending on whether the carbon oxidizes under first order kinetics control or diffusion control (row 2 column 3). It can be shown that $C_{ch,p} = 4 \pi a (\rho D)_{eff}$ [26] (row 1, column 3, Table 3; see also Chapter 9 [29]) under diffusion control. The *G#* for carbon clouds oxidizing under first order kinetics was shown to be equal to $\psi_T^2$ where $\psi_T$ is called Thielemodulusin carbon combustion literature. When G increases to incipient group number $G_{inc}$, at which the core contains almost no oxygen (or oxygen concentration reaches almost "extinction" level), the carbon oxidation near the core of the cloud ceases.

Gilot et al. [30] conducted experimental studies on the oxidation of black soot particles using a thermo-gravimetric analyzer (TGA) where a known amount of the sample (in mg) is exposed to an oxidizing atmosphere with a high temperature source of 600–900 °C. Mass versus time was measured. The specific combustion rate (g/s per g mass of cloud), or SERR, was found to decrease as the sample mass (or cloud size) is increased validating OD combustion model. Figure 4 shows the results with oxygen concentration as a parameter. The authors attribute the decreasing values of specific combustion rates to the limited diffusion of oxygen, which is consistent with the OD/GC modeling of dense clouds where the consumption rate of $O_2$, and hence the carbon burn rate (or $O_2$ consumption rate), decreases with increased sample or cloud mass.

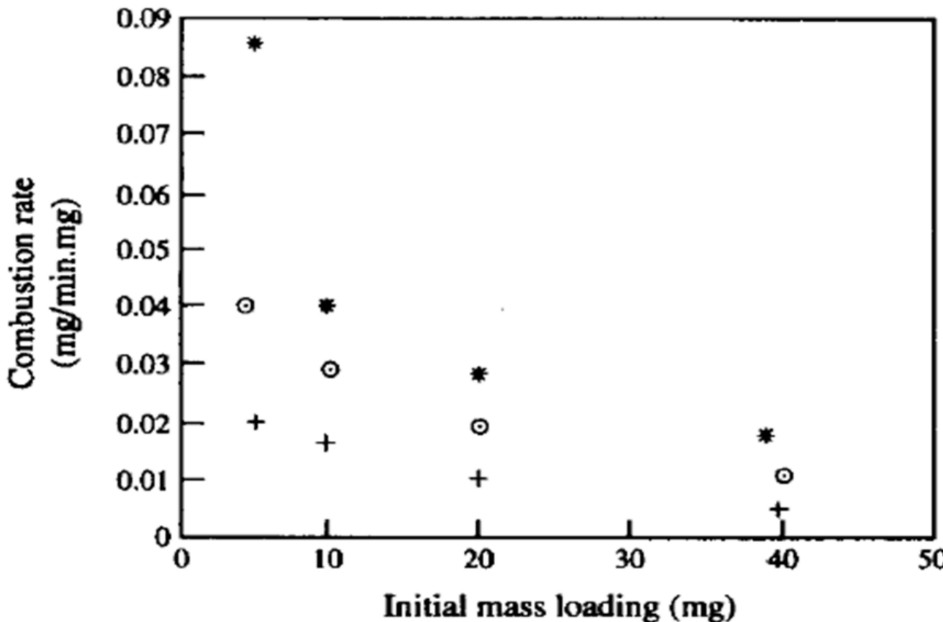

**Figure 4.** The effect of mass loading and oxygen concentration on the specific combustion rate (SCR) of black soot particles. SERR ∝ SCR. Oxygen effect at 600 °C +: 5% (volume): ⊙ 10% (600 °C) and ✷: 15% $O_2$ (adopted from [30]).

## 2.3. Oxidation of Nutrients in BS

The macronutrients are typically CH and F. Oxygen, and water are rarely considered as nutrients [31]. Trayhurn, in his recent review [32] sets forth arguments that these are

also nutrients, particularly when there is severe oxygen deficiency. However, water and oxygen are excluded as nutrients in the current review.

Just like oxidation of C(S) which occurs in two steps (Section 2.1), the oxidation of CH occurs in two steps: the first step is glycolysis of CH to pyruvate (an anaerobic process) outside of the mitochondria within cytoplasm of cells [33] and the second step is oxidation of pyruvate within mitochondria ((MITO, aerobic process). The benign cells derive 88% of energy from oxidation in Mito and 12% from glycolysis [34,35].

The oxygen in MITO serves as an electron acceptor during oxidation.

ET (electron moles transferred per mole fuel) = 4 × (stoichiometric oxygen moles per mole fuel), assuming C to $CO_2$, H to $H_2O$, and S to $SO_2$ for oxidation of $C_cH_hN_nO_oS_s$ fuels. For Glucose, ET = 24. As such, energy per electron mole is given by $(1/4)*(HV_{O2}$ in kJ/mole of $O_2$). Thus, the energy released per mole of electron for any C–H–O fuel is 454/4 = 115 kJ/mole of electron which is close to 111.1 as reported in [15]. The overall CH oxidation reaction is given as

$$C_6H_{12}O_6 + 6\ O_2 + 36\ ADP + 36\ Pi + 6\ O_2 \rightarrow 6\ CO_2 + 6\ H_2O + 36\ ATP,$$

On mass basis (excluding ATP, ADP, Pi),

$$0.94\text{g of } C_6H_{12}O_6 + 1\text{ g of } O_2 \rightarrow 1.38\text{ g } CO_2 + 0.56\text{ g } H_2O$$

Ref. [34] reports only 34 ATP per glucose molecule, and benign cells derive 88% of energy from oxidation and 12% from glycolysis [35]. On the other hand, cancer cells rely on glycolysis to provide most of the energy.

Energy Release Pathways: Depending on the local oxygen concentration and energy required, the cells can release energy through two pathways: (1). Oxidative phosphorylation (OXPHOS or Citric acid or Krebs cycle) is promoted by normal cells in the presence of "enough" oxygen level, which generate 36 ATP per glucose molecule. (2). The inefficient and more acidic glycolysis pathway to lactic acid is promoted when $O_2$ levels are deficient and generate only 2 ATP per glucose molecule resulting in a high lactate production [36], i.e., there is an augmentation of anaerobic glycolysis under low $p_{O2}$. This is known as the Pasteur effects [37] where glucose consumption is increased in hypoxic conditions or when tumor or virus cells are present. The ERR is only about 12% of the energy from oxidation. The absence of mitochondria in cells (e.g., erythrocytes) can also lead to anaerobic glycolysis as a major pathway for energy release.

### 2.4. Oxidation Models for Organs

Each organ contains billions of cells with a specific vital function, and they are irrigated by capillaries (micro-vasculature) with nutrients and $O_2$.

Oxidation Models for Organs: Oxidation of CH within each organ is modeled as though an organ is made up of a group of multiple cylinders of radii, R (Figure 5a) that contain metabolic cells supplied with oxygen from a capillary of radius $r_{cap}$ at $Y_{O2cap}$ located at the axis of each cylinder (Figure 5b). This model is known as the Krogh Cylinder model [38] and could be re-termed as the capillary on axis (COA) model. For a detailed review of Krogh models, see Ref. [39]. Each cylinder contains metabolic cells immersed in interstitial fluid (IF, the bulk phase in the cell cloud) (Figure 5c). Typically, the mixture in capillaries is CH rich and the oxidation is essentially controlled by diffusion of dissolved $O_2$ from capillaries to the IF and by the kinetics of oxidation within cells. Under kinetics control, Michaelis Menten kinetics are used for oxidation within MITO: $C_{Ch,k}$, is known for each organ; the constants depend on Michaelis Menten (MM) constants $k_{MM}$, in kinetics expression.

$$\dot{w}_{O2,cell} = \dot{w}_{O2,cell,max}\left(\frac{Y_{O2}}{Y_{O2}+k_{MM}}\right), \quad \dot{w}_{O2,cell} \approx C_{Ch,cell,kin}\ Y_{O2}$$
$$\text{when } Y_{O2} << k_{MM},\ C_{Ch,cell} = C_{Ch,cell,kin} = \left(\frac{\dot{w}_{O2,cell,max}}{k_{MM}}\right) \tag{16}$$

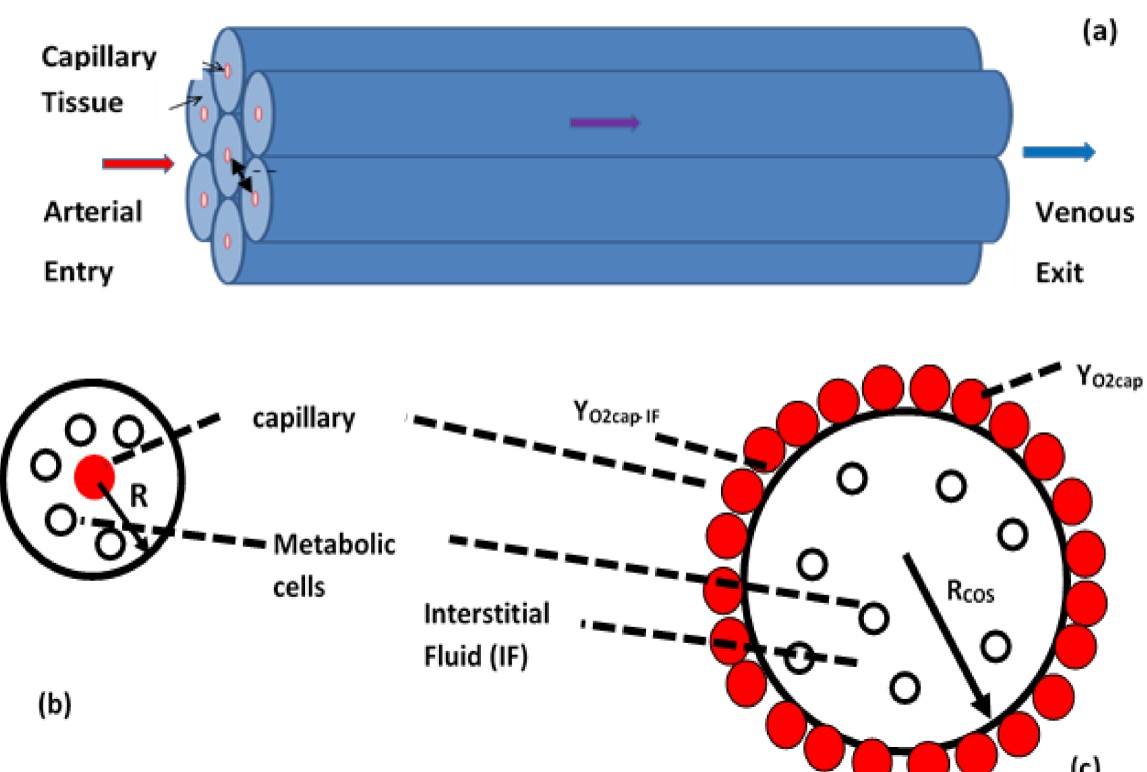

**Figure 5.** (**a**) Model: Illustration of Multi-Krogh cylinders, each consisting of a capillary on axis (COA) at $Y_{O2cap}$, (**b**) Capillary on Axis (COA) of radius $r_{cap}$ on axis at $Y_{O2cap}$ with diffusion of $O_2$ away from the axis towards the impermeable surface of radius R, (**c**) Capillary on Surface (COS) where the surface is covered with capillaries each of radius $r_{cap}$. The diffusion of oxygen is away from the surface towards the core of cylinder.

Thus, first order kinetics is followed when $Y_{O2}<<k_{MM}$ (or under OD condition). The model is like the "rich flammability limit" model in combustion science where the $O_2$ conservation equation is used to predict the rich flammability limit [40]. In addition to COA models, there are solid cylinder models where oxygen diffuses from the surface towards the core [41]. This has been re-termed as the capillaries on surface (COS) (Figure 5c) model.

The interface between IF and capillaries is at mass fraction of oxygen $Y_{O2cap-IF}$ and will be the same as $Y_{O2cap}$ if the mass transfer across capillaries is very fast. The COS model, or solid cylinder model in biology, is similar to the carbon dust cloud model used extensively in combustion science (see Figure 3) [26]. The cells of density, $\rho_{Cell}$, replace the density of carbon particles and the density of IF (density $\rho_{IF}$ which is almost the same as $\rho_{Cel}$)) replaces the density of the bulk gas phase in the cloud combustion model. The $Y_{O2, cl}$ at the carbon cloud surface (see Figure 3) is replaced by $Y_{O2cap-IF}$. In addition to the cylinder model, slab and spherical geometries are also considered for modeling the metabolism within an organ [42].

### 2.5. Oxygen Deficient Metabolism (ODM) in Organs

When an efficient vascular bed is absent, hypoxia occurs. The brain's mass is only 2% of the total body mass but consumes 25% of the total oxygen for production of ATP for electrical activities [43] indicating high consumption rate of $O_2$ per unit mass of brain. More than 30% of COVID-19 patients have faced "brain fog" and impaired cognition. One of the major causes for this is a lack of oxygen or OD [44]. The tissue oxygenation level (typically $p_{O2}$ = 40–50 mm Hg, in some regions of the brain and as low as 16 mm Hg [32]) is hypoxic compared to alveoli levels ($p_{A,O2}$ = 104 mm Hg). Further, 50% saturation in arterial blood from lungs of COVID patients leads to an extremely rich mixture of CH and $O_2$, and the glycolysis pathway is promoted [45] under OD. About 44% of those recovered from COVID

face neurological and psychiatric illnesses including delirium, seizures, encephalitis, etc. The studies by NYU School of Medicine found that the main culprit is low oxygen levels or OD in the organs of the body for extended periods of time [46].

The diffusional distance from the capillary for oxygen is about 100–200 μm. Those cells far from the capillary (e.g., the core of the cylinder in the COS model (Figure 5c) or the cells near the surface of the cylinder in COA model (Figure 5b)) may not receive oxygen, resulting in the cessation of oxidation. When the oxygen mass fraction falls below lethal levels, $Y_{O2, leth}$ (somewhat like limiting oxygen index in combustion science), the oxidation reaction ceases, and cells rely more on glycolysis [37] for energy which generates more cell building ingredients. Thus, poorly oxygenated tissue or lethal parts of the tissue undergo glycolysis with an end product of lactic acid, which may lead to the destruction of healthy cells [47,48]. According to the Warburg hypothesis in biology, they also lead to the partial destruction of normal cells due to a reduced energy release and the creation of cancer cells [48].

Singer et al. are probably the first in the field of biology to postulate the role of OD, or "crowding effect," on the metabolic rates of in vitro (test tube) samples. They developed a phenomenological type of model [49] to explain the decrease of $SMR_k$ with an increase in size of the in vitro sample. The model assumes (a) a spherical geometry of radius R, (b) a surface exposed to an $O_2$ rich atmosphere (similar to COS model), (c) a thin oxygenated (or aerobic) shell of thickness, δ (about 100 μm) near the surface, (d) a uniform source (US) model for $O_2$ consumption within the aerobic shell (R-δ < r < R) and (e) the anaerobic core (0 < r < R-δ) undergoing glycolysis. When the sample mass is small (m << 0.2g, R < δ), it is fully aerobic with $Y_{O2}(r) \approx Y_{O2cap-IF}$ everywhere; when the sample is large (about 0.2 g and higher or when R > δ), oxidation occurs at the outer shell with the core supplying only 5–10% of the oxidative energy release via glycolysis. If δ is assumed to be constant, then the aerobic volume fraction decreases for larger samples, which results in a lower SMR and validates the model with experimental data (Figure 6) [25] from in vitro samples.

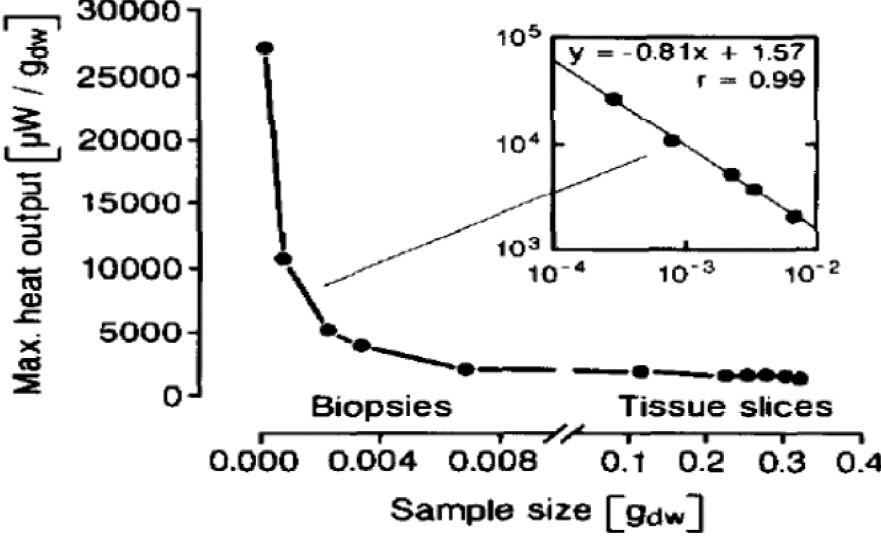

**Figure 6.** Maximum energy release rate (maximum ERR in microwatts) per g of dry weight (dw) versus the sample size of the rat liver. For biopsies, a sample mass of the order 2 mg has a higher SMR compared to the sample mass of 0.2 g. Note the insert which shows a linear curve, fit to limited mass range. Adopted from Singer et al. [6]. See Figure 4 for similar results in combustion science.

### 2.6. Organ Mass Based Allometry (OMA)

It is apparent from the review of GC/OD literature from both fields that the SEER rates vary with cloud mass. However, the biology literature expresses the variation of SEER of organs (equivalent to the multiples of cell clouds) in terms of body mass using empirical allometric constants (Equation (7)). In order to compare these results with those

from combustion science, the BMA for SERR of organs must be converted into organ mass based allometry (OMA) so that the allometric exponents can be interpreted in terms of OD deficient metabolism. Thus, eliminating $m_B$ between Equations (7) and (8) [20]

$$\dot{q}_{k,m'}\left\{\frac{\text{Watts}}{\text{kg of organ k}}\right\} = E_k \; m_k^{\;F_k}, \quad F_k = \left\{\frac{f_k}{d_k}\right\}, \quad E_k = \left\{\frac{e_k}{c_k^{\left(\frac{f_k}{d_k}\right)}}\right\}, \quad k = K, H, Br, L, R \qquad (17)$$

See Table 2 for $F_k$ values based on 6 species and these will be re-presented in Section 5.5. It will be seen in Sections 5.5 and 5.6 that the OD plays a major role in the allometric laws for the metabolic rates of vital organs and the whole body.

### 3. Objectives

The objectives of the current work are as follows: (i) extend the literature from OD combustion to OD metabolism including the transfer of the *G* number from combustion literature to $G_{OD}$ number of each organ where OD represents oxygen deficiency, (ii) show that the local curve fitting of effectiveness factor ($\eta_{\text{eff}, k}$) vs. $G_{OD, k}$ of an organ *k* leads to OMA and yields $F_k$ values vs. $G_{OD,k}$, (iii) explain the negative exponents $f_k$ and $F_k$ of organs and show that $-\frac{1}{3} < F_k < 0$, (iv) validate the inequality with the experimental data for 6 and 111 species, (vi) explain and show that the negative exponents $b'$ of the whole body is affected by the size of the organ (or size of cell clouds) within BS and (vii) present the incipient $G_{OD,INC}$ for organ k at which oxygen concentration reaches a lethal concentration in the core of the cell cloud resulting in a cessation of oxidation and the inception of glycolysis, a precursor to the creation of cancer cells. While a hypoxic condition for the whole body is given in terms of a saturation percentage in the arterial blood ($\approx$92%) and (Hb) levels in blood, the manuscript presents $G_{OD \, Inc}$ at the organ level, which is an indication for the onset of the hypoxic condition of an organ.

### 4. Materials and Methods

*Governing Equations and Boundary Conditions for the ODM Model of Cell Clouds*

Consider the spherical cloud of radius R and mass $m_{cl}$, containing n metabolic cells per unit volume of interstitial fluid (IF), with the cell cloud surface at $Y_{O2cap-IF}$ (Figure 7).

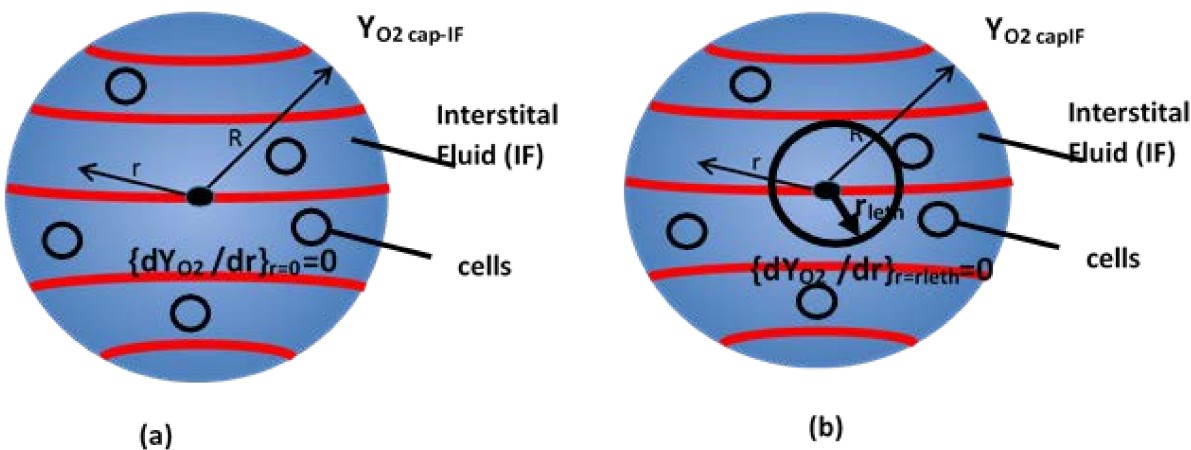

**Figure 7.** Illustrations of the COS-$O_2$ model in spherical geometry with the solid lines representing the capillaries. (**a**) No lethal volume, (**b**) lethal volume near core. Under poor oxygenation such as in COVID-19 where saturation percentage is as low as 50%, lethal volume percentage is much higher compared to a normal healthy person.

Column 4, Table 3 presents the conservation equations, boundary conditions with and without lethal volume, and solutions for $O_2$ profiles. The spherical cell cloud mass, $m_{cl}$, of an organ *k* is assumed to be proportional to the organ's mass, $m_k$, since an organ is

modeled as a cluster of spherical cell clouds. The relation for oxygen consumption rate of a cell ($\dot{w}_{O2,cell}$) located at r is given in row 1, column 4 of Table 3.

**Table 3.** Conservation Equations, Boundary Conditions, G number, Oxygen Profiles and Effectiveness Factors for Carbon Dust Clouds in Combustion Literature vs. Cell Clouds in Biology—Spherical Geometry.

| # | Geometry | Carbon Dust Cloud | Cell Clouds-COS-$O_2$ |
|---|---|---|---|
| 1 | $O_2$ consumption rate per particle or per cell | $\dot{w}_{O2,p}\left(\frac{g}{s}\right) = C_{ch,p}\ Y_{O2}$ | $\dot{w}_{O2,cell}\left(\frac{g}{s}\right) = C_{ch,cell}\ Y_{O2}$ |
| 2 | Characteristic $O_2$ consumption rate per particle ($C_{ch,p}$) or Characteristic $O_2$ consumption rate per cell ($C_{ch,cell}$) | $C_{ch,p} = C_{diff,p}\left(\frac{g}{s}\right) \approx\ 2\pi\ \rho D\ d_p$ under diff.control<br><br>$C_{ch,p} = C_{ch,p,kin}\left(\frac{g}{s}\right) = \frac{\dot{w}_{O2,p,max}}{k_{LM}}$,<br>$k_{LM}$, Langmuir Constant<br>$Y_{O2} << k_{LM}$<br>$\dot{w}_{O2,p,max} = \dot{w}''_{O2,p,max}\pi d_p{}^2$<br>First order Langmuir (LM) kinetics | $C_{ch,cell} = C_{diff,cell}\left(\frac{g}{s}\right) \approx\ 2\pi\ (\rho D)_{eff}\ d_{cell}$ under diff.control<br><br>$C_{ch,cell} = C_{ch,cell,kin}\left(\frac{g}{s}\right) = \frac{\dot{w}_{O2,cell,max}}{k_{MM}}$, $Y_{O2} << k_{MM}$<br>$\dot{w}_{O2,cell,max} = \dot{w}'''_{O2,cell,max}\frac{\pi d_{cell}{}^3}{6}$<br>First order Michaelis Menten (MM) kinetics |
| 3 | Bulk phase within cloud | Gas of very low density (order of 1 g/cm$^3$) compared to carbon particles of density 1300 g/cm$^3$. | Interstitial fluid (IF) with similar densities for cell and fluid |
| 4 | Conservation Equations (dimensional) | $(\rho D)_{eff}\frac{1}{r^2}\frac{d}{dr}\left(r^2\frac{dY_{O2}}{dr}\right) =\ \dot{w}'''_{O2}(r)$<br>$\dot{w}'''_{O2}e(r) =\ n\ \dot{w}_{O2,p}(r)$ | $(\rho D)_{eff}\frac{1}{r^2}\frac{d}{dr}\left(r^2\frac{dY_{O2}}{dr}\right) =\ \dot{w}'''_{O2}(r)$<br>$\dot{w}'''_{O2}(r) =\ n\ \dot{w}_{O2,cell}(r)$ |
| 5 | Conservation Equations (non-dimensional) | $\frac{1}{\xi^2}\frac{d}{d\xi}\left(\xi^2\frac{dY_{O2}}{d\xi}\right) =\ G\ Y_{O2}, \xi\ =\ \frac{r}{R}$ | $\frac{1}{\xi^2}\frac{d}{d\xi}\left(\xi^2\frac{dY_{O2}}{d\xi}\right) =\ G_{OD}\ Y_{O2}, \xi\ =\ \frac{r}{R}$ |
| 6 | Dimensionless $G$ number | $G\ =\ \frac{C_{ch,p}\ n\ R^2}{(\rho D)_{eff}}, G = \Psi_T{}^2$, first order kinetics | $G_{OD}\ =\ \frac{C_{ch,cell}\ n\ R^2}{(\rho D)_{eff}}, G_{OD} = \Psi_{OD,T}{}^2$, first order kinetics |
| 7 | Boundary Conditions with lethal volume | - | $Y_{O2} = Y_{O2,\text{cap-IF}}$ at $r = R$  $dY_{O2}/dr = 0$ at $r = r_{leth}$ (Figure 7b) |
| 8 | Boundary Conditions without lethal volume | $Y_{O2} = Y_{O2,\text{cl}}$ at $r = R$<br>$dY_{O2}/dr = 0$ at $r = 0$ | $Y_{O2} = Y_{O2,\text{cap-IF}}$ at $r = R$<br>$dY_{O2}/dr = 0$ at $r = 0$ (Figure 7a) |
| 9 | Oxygen Profiles with lethal Volume, $\frac{Y_{O2}(\xi)}{(Y_{O2})_{\xi=1}}$, | - | See Appendix A (Equations (A2) and (A3)) |
| 10 | Oxygen Profiles without lethal Volume, $\frac{Y_{O2}(\xi)}{(Y_{O2})_{\xi=1}}$, [26] | $\left(\frac{1}{\xi}\right)\frac{\text{Sinh}\left(G^{1/2}\xi\right)}{\text{Sinh}\left(G^{1/2}\right)}$ | $\left(\frac{1}{\xi}\right)\frac{\text{Sinh}\left(G_{OD}{}^{1/2}\xi\right)}{\text{Sinh}\left(G_{OD}{}^{1/2}\right)}$ |
| 11 | Oxygen Mass Fraction at core {without lethal volume}, $\frac{Y_{O2}(0)}{(Y_{O2})_{\xi=1}}$, [26] | $\frac{1}{\text{Sinh}\left(G^{1/2}\right)}$ | $\frac{1}{\text{Sinh}\left(G_{OD}{}^{1/2}\right)}$ |
| 12 | Incipient Group/OD Combustion [1] | $Y_{O2,0}\rightarrow Y_{O2ext,}$ solve for<br>$G_{inc}\ G_{inc} = \left[\text{Sinh}^{-1}\left\{\frac{1}{\left(\frac{Y_{O2ext}}{Y_{O2,cl}}\right)}\right\}\right]^2$ | $Y_{O2,0}\rightarrow Y_{O2,leth,}$ solve for<br>$G_{OD,inc}\ G_{OD,inc} = \left[\text{Sinh}^{-1}\left\{\frac{1}{\left(\frac{Y_{O2leth}}{Y_{O2,cl}}\right)}\right\}\right]^2$ |
| 13 | Effectiveness factor, $\eta_{eff} = (Y_{O2avg}/Y_{O2cap\text{-}IF})$ at G, [26], without lethal volume | $\frac{3}{\sqrt{G}}\left\{\frac{1}{\tanh(\sqrt{G})} - \frac{1}{\sqrt{G}}\right\}$, 1-(G/15)→1 as G→0 $\frac{3}{\sqrt{G}}$,<br>SERR $\propto m_{cl}{}^F$, $F_k\rightarrow -1/3$, $G > 100$ | $\frac{3}{\sqrt{G_{OD}}}\left\{\frac{1}{\tanh(\sqrt{G_{OD}})} - \frac{1}{\sqrt{G_{OD}}}\right\}$ SERR $\propto m_k{}^{Fk}$ {biology},<br>$F_k\rightarrow 0$ (isometric law) as G→0 $\eta_{eff}\rightarrow\frac{3}{\sqrt{G_{OD}}}$, $F_k\rightarrow$<br>$-1/3$, $G_{OD} > 100$ |

[1] The transfer of the incipient combustion number; $G_{INC}$, from combustion to incipient number; $G_{OD,INC}$, for organs in biology at which cells at core become oxygen deficient; oxidation reaction ceases. The fermentation process, a precursor to the creation of cancer cells, occurs for cells near the core of cell clouds.

The cell within IF consumes oxygen either under diffusion control ($O_2$ transport rate << kinetics rate) or under first order kinetics control (kinetics rate << $O_2$ transport rate} (equations in row 1, column 4, Table 3). For example, $C_{ch,cell} = 2\pi\ d_{cell}\ (\rho D)_{eff}$ [26] under diffusion control following the OD combustion literature. See Equation (16) for kinetics control. In both cases, the oxidation rate for the cell ($\dot{w}_{O2,cell}(r)$) is proportional to the local oxygen mass fraction, $Y_{O2}(r)$. Thus, the term $C_{Ch,cell}$ changes when a cell metabolizes under kinetics control or diffusion control. The $O_2$ consumption rate per unit volume,

$\dot{w}_{O2}'''$ (r), is given as n $\dot{w}_{O2,cell}${row 4}. The conservation equations for oxygen species are given in dimensional form (row 4). The boundary conditions are given in row 7 with lethal volume, row 8 without lethal volume [26]; the $(\rho D)_{eff}$, is called the effective mass diffusivity of $O_2$, which accounts for the presence of cells within the cloud.

**Non-Dimensional $G_{OD}$ Number for Organs and Physical Meaning**: Following the combustion literature, the group number $G_{OD}$ of cell clouds is defined as,

$$G_{OD,k} = \left\{ \frac{C_{ch,cell} \, n \, R^2}{(\rho D)_{eff}} \right\}_k = \frac{\text{Charactristic } O_2 \text{ consumption rate by all cells within cell loud}}{\text{Charactristic } O_2 \text{ diffusion rate to the cells from capillaries}}, \quad (18)$$

The $G_{OD}$ # is referred to as the OD number (row 6, column 3). When constant of proportionality $C_{Ch,cell}$ is selected from the expression for kinetics-controlled metabolism, $G_{OD}$ is same as (Thiele Modulus)$^2$. Higher $G_{OD}$ numbers result in a higher oxygen deficiency within the cell cloud and the more the hypoxic condition of the organ, the more likely the cells are to adopt a glycolytic pathway for energy release. Both G and $G_{OD}$ are proportional to $R^2 \propto m_{cl}^{2/3}$ for dust clouds or $m_k^{2/3}$ for organ *k*. The conservation equations for oxygen species are given in non-dimensional form (row 5, non-dimensional radius, $\xi = r/R$) for dust clouds (column 3) and cell clouds (column 4) while row 6 defines the G# for dust clouds and $G_{OD}$ # for applications to cell clouds in biology.

## 5. Results

### 5.1. Equilibrium Levels of Oxygen in Arterial Blood

The boundary condition for solution of conservation equations given in Table 3 require a knowledge of $Y_{O2,cap}$. The mass fractions of oxygen in capillaries of an organ *k* depends on the equilibrium mass fractions of oxygen ($Y_{O2,cap} \approx Y_{O2,a}$) in the arterial blood. Hence the equilibrium levels of oxygen within arterial blood at partial pressure, $p_{a,O2}$, will be presented first. The $Y_{O2,a}$ consists of two parts: dissolved $O_2$ and $O_2$ present as OHb. Based on the plot shown in Figure 2 (saturation % vs. $p_{O2}$, adopted and modified from ref. [12]) and data presented in the caption of Figure 2 shows the dissolved oxygen content in arterial blood can be plotted as a function of $p_{O2}$ ($\approx p_{A,O2}$). The results for both the total and dissolved oxygen contents are shown in Figure 8. In COVID-19 patients, Hb denaturation occurs, thus reducing available $(Hb)_{in}$ for oxidation to Hb $(O_2)_{in}$ [45] and thus the transfer of oxygen across alveoli membrane to the capillary is disrupted, sometimes as low as 50% saturation (or saturation fraction = 0.5) of arterial blood. It also affects the transfer of $CO_2$ from blood to alveoli. The normal pH for humans is 7.35–7.45 and $CO_2$ loading reduces pH to 7.25, which increases the breathing rate.

### 5.2. Oxygen Profiles in Cell Clouds

Appendix A presents the solution for oxygen concentrations when the lethal volume is finite (Equations (A2) and (A3)). When $\xi_{leth}$ is set to zero in Equation (A2), one recovers the solution for $Y_{O2}$ without lethal volume. The non-dimensional solutions for the oxygen mass fraction for carbon clouds and cell clouds are given in row 9 and 10 for cell clouds While $O_2$ mass fraction ranges around 300 ppm for healthy humans, the total $Y_{O2}$ could be as low 150 ppm and dissolved $O_2$ as low as 1.5 ppm at 50% saturation (Figure 8) for COVID-19 patients, indicating a very rich mixture of CH and $O_2$. Low $O_2$ concentration results in an increase of lethal volume within each vital organ. Detailed solutions for metabolic rates and extent of lethal volumes have been obtained for cell clouds involving slab, cylinder, and spherical geometries. However, the subsequent sections deal with presentations of solutions of a spherical cloud without lethal volume so that the methodology for deriving allometric laws of organs from effectiveness factor charts and validation of Kleiber's law for the whole body can be kept in a tractable form. Figure 9 presents $O_2$ profiles with *G* (carbon clouds) or $G_{OD}$ (cell clouds) as parameters. At low $G_{OD}$, the profile is almost flat, indicating a uniform $O_2$ consumption. They become steeper at high $G_{OD}$, indicating an aerobic shell of thickness $\delta$ near r = R while most of the inner cloud is anerobic.

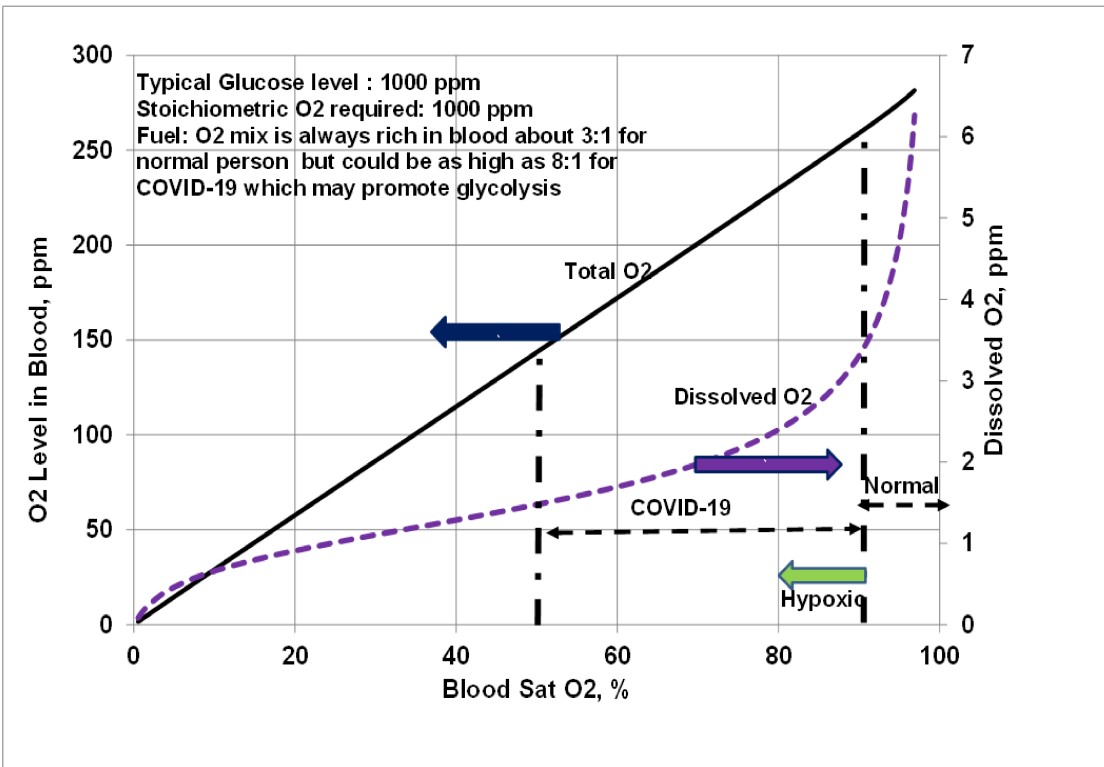

**Figure 8.** Oxygen contents in the blood versus the saturation % in arterial blood in ppm or g per million g of blood. Saturation % for COVID patients range from 50–90%. It is apparent that COVID-19 patients receive less oxygen in their blood at 50% saturation, and particularly dissolved blood, which is transferred to mitochondria resulting in concentration as low as about 2 ppm, leading to oxygen deficiency. Further, the blood becomes richer in glucose, thus cells do not receive an adequate oxygen supply, resulting in a reduced ATP production per cell.

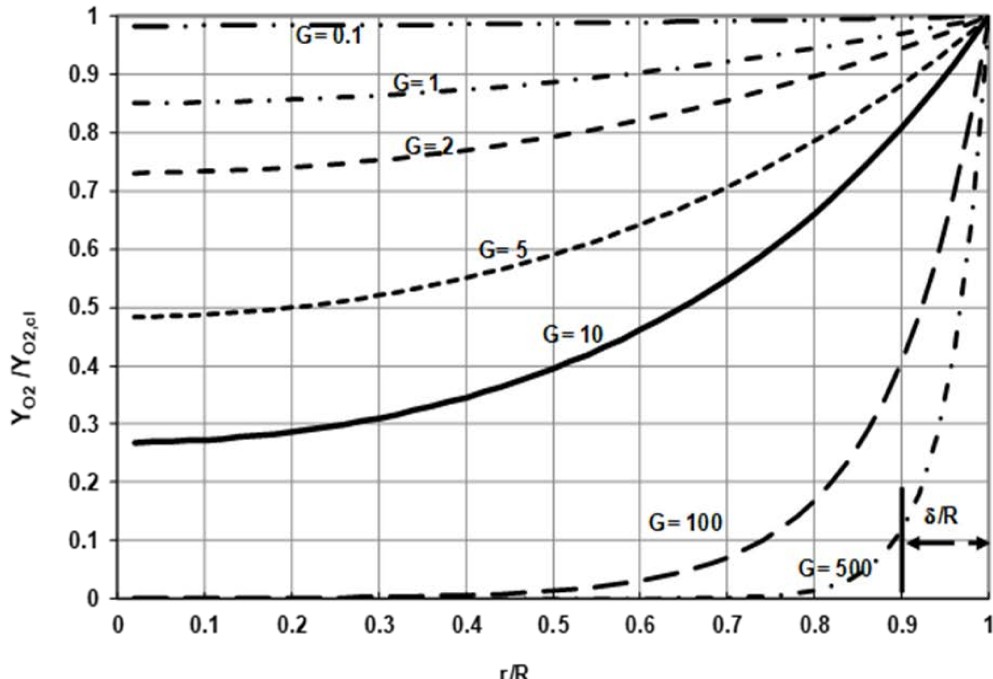

**Figure 9.** Oxygen profiles for a spherical cloud of carbon particles or spherical group of carbon particles of radius R; these are the same as those for porous char pf particle of radius R. For biological applications, with spherical geometry, change $Y_{O2}$ at cloud surface to $Y_{O2cap}$-IF and $G$ to $G_{OD}$. At high $G_{OD}$ number, aerobic thickness is also shown. For cell clouds, change $G$ to $G_{OD}$.

### 5.3. Effectiveness Factor ($\eta_{eff}$)

Adopting the methodology used in combustion literature (e.g., porous char combustion), the results for $O_2$ profiles are used to determine the $O_2$ consumption rate for whole spherical clouds of radius R:

$$\dot{w}_{O2} = n \int_0^R C_{ch,cell} \, Y_{O2}(r) \, 4\pi r^2 \, dr, \quad \text{Cell Cloud} \tag{19}$$

Defining the effectiveness factor, $\eta_{eff}$ as [26,50]:

$$\eta_{eff} = \frac{Y_{O2,avg}}{Y_{O2,cap-IF}} = \frac{O_2 \text{ consumption rate by all cells within cell cloud with } Y_{O2}(r)}{O_2 \text{ consumption rate by all cells within cell cloud with each cell at } Y_{O2} = Y_{O2,cap-IF}} \tag{20}$$

The numerator is proportional to the actual consumption rate, while the denominator is proportional to the hypothetical consumption rate when $Y_{O2} = Y_{O2, cap-IF}$ for all cells. Note that $\eta_{eff}$ is only a function of $G_{OD}$ for COS-$O_2$ models. Solutions for $\eta_{eff}$ are given in row 13 of Table 3. The solutions for $\eta_{eff}$ for the three geometries of cell clouds are the same as those from dust cloud literature [50]. Using results from combustion literature, Figure 10 presents the variation of $\eta_{eff}$ with $G_{OD}$ for the three geometries (See also equation in Table 3, row # 13). It is seen that effectiveness factors are close to each other under identical surface area to volume (S/V) ratios or similar surface area of capillaries to mass ratio, $S_{cap,m}$ within organs.

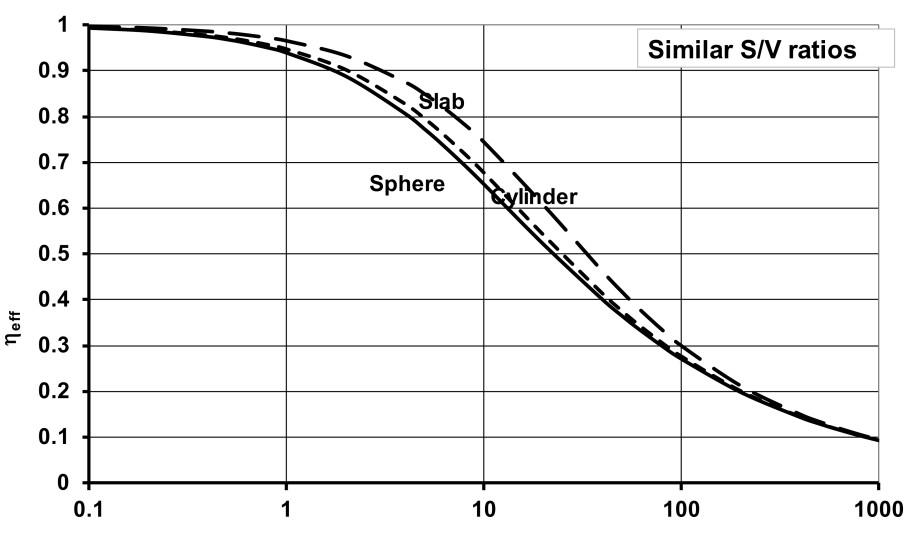

**Figure 10.** Effectiveness Factor of organ k vs. G (or $G_{OD, k}$) for same external surface area to volume ratio (S/V) or similar $S_{cap, m}$ (surface area to mass ratio)} when organ *k* is assumed to consist of multiple spheres, cylinders, and slabs. See also [50]. COS-$O_2$ model and no lethal volume.

Knowing $G_{OD}$, the $\eta_{eff}$ is determined from Figure 10 and the oxygen consumption rate and hence the $SMR_k$ oof any organ k can be estimated using Equation (21):

$$SMR_k \left(\frac{W}{kg}\right) = \frac{\dot{w}_{O2,cloud} \, HV_{O2}}{m_{cloud}} = \dot{w}_{O2,m} \, HV_{O2} = \eta_{eff,k} \, \frac{C_{ch,cell} \, Y_{O2,cap-IF} \, n}{\rho} HV_{O2} \tag{21}$$

### 5.4. Allometric Laws Derived from Effectiveness Factor Charts

The $G_{OD}$ changes depending on the size of organ $k$ of mass $m_k$ and reactivity of cells within organ $k$. Large organs are typically at a high $G_{OD}$ and small organs are at low $G_{OD}$. Thus, the solution for $\eta_{eff}$ vs. $G$ or $G_{OD,k}$ presented in Table 3 (row # 13) can also be presented in the form of organ mass based allometric law using the following procedure.

For small intervals of $G_{OD}$ (Figure 10) of any organ k, one can set:

$$\eta_{eff,k} \ = \ A_k \ G_{OD}{}^{s_k}, k = Br, H, K, L \tag{22}$$

and determine the $s_k$ values by curve fitting $\eta_{eff}$ vs. $G_{OD,k}$ and treating $s_k$ as a constant for a narrow range of $G_{OD,\,k}$. Differentiating Equation (22) with $G_{OD,k}$ (Figure 10),

$$\frac{d\eta_{eff,k}}{dG_{OD,k}} \ = \ \frac{s_k}{G_{OD,k}}, k = Br, H, K, L \tag{23}$$

where $\frac{d\eta_{eff,k}}{dG_{OD,k}}$ is obtained using the relation given for $\eta_{eff,\,k}$, with $G_{OD,\,k}$ (row 13, Table 3). Using the result in Equation (23), the $s_k$ values can be determined, plotting the results for $s_k$ vs. $G_{OD,\,k}$. (Figure 11). Note that Figure 10 and hence Figure 12 are based on COS-$O_2$ model. The $G'_{OD,\,k} \propto m_k{}^{(2/3)}$, and hence $\dot{q}_{k,m} \propto \eta_{eff,k} \propto G_{OD,k}{}^{s_k} \propto m_k{}^{(2/3)s_k}$. Comparing this with Equation (17),

$$F_k \ = \ \frac{2}{3} \, s_k, \ k = Br, H, K, L, \ COS\text{-}O_2 \ \text{model} \tag{24}$$

The variation of $F_k$ with $G_{OD,\,k}$ is shown in Figure 11. All the $F_k$ values are negative. As $G_{OD,k} \to 0$ (small organ), $F_k \to 0$ and as $G_{OD,k} \to \infty$ (large organs), $F_k \to (-1/3)$. Hence, the $F_k$ values satisfy the bounds: $-\frac{1}{3} < F_k < 0$ for COS-$O_2$ models. While the phenomenological model of Singer with aerobic film of constant thickness and anaerobic core or lethal core [49] is used to explain the decrease of SMRk with an increase in size of the in vitro sample (equivalent to negative values for "$f_k$" coefficients used in the body mass based allometry), the current model assumes oxidation proceeding even at very low oxygen mass fraction within core, but still yields a decrease of SMR with an increase in size of the organ, More notably, it yields the bounds on "Fk" coefficients used in the organ mass based allometry

### 5.5. Validation with Experimental Data for Allometric Exponents

Effect of mass of organ-6 species: The organ masses were normalized with each kidney (K) mass (smallest mass). Using Equation (8), one can obtain normalized an allometric relation for $m_k{}^*$ which is a ratio of the organ mass to kidney mass.

$$m_k* = \ \frac{m_k}{m_K} = \ \left( \frac{c_{k,6}}{c_{K,6}} \right) m_B{}^{d_{k,6}-d_{K,6}} = \ g_{k,6} m_B{}^{h_{k,6}}, \ g_{k,6} = \left( \frac{c_{k,6}}{c_{K,6}} \right), \ h_{k,6} = d_{k,6} - d_{K,6} \tag{25}$$

where $c_{k,\,6}$ is the $c_k$ value for organ k based on six species. Table 4 presents the values of $g_{k,\,6}$ and $h_{k,\,6}$. The $m_{Br}{}^*$ ranges from 3.3–2.1, $m_H{}^*$ from 1.6 (0.48 kg rat) to 3.1 (65 kg human), and $m_L{}^*$ from 9.3–10.3, (row 3, Table 4) indicating a narrow variation, whereas the mass of the body ranged by a factor of 150. This is also apparent from Equation (25) where the values of $|h_{k,\,6}|$ are very low, and hence non-dimensional masses $m_k{}^*$'s are (row 3, Table 4) almost insensitive to variation in body mass.

Figure 12 shows the variation of the experimental exponent $F_k$ with the variation of mk* for the six species (average value within square filled pattern). It is noted that the larger the organ mass, the more negative the value for $F_k$ (indicating a higher OD). Based on the mean, the liver is larger with $F_k$ near the theoretical limit of $-0.33$, while $F_k$ is near zero for kidneys of smaller mass following the isometric law. Smaller organs are known to have faster $O_2$ delivery systems to the cells [23] since $O_2$ gradients are steeper. The metabolism of smaller organs, and hence smaller animals, may approach the maximum possible metabolic rate and thus they exhibit a lower tolerance to any environmental fluctuations.

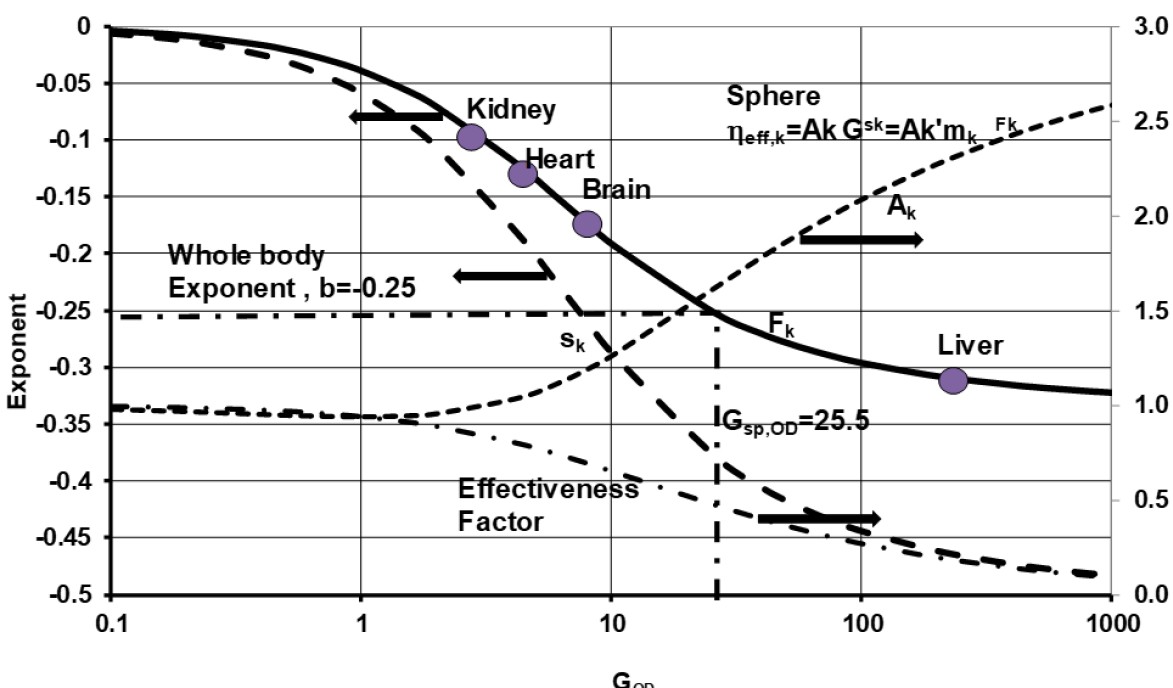

**Figure 11.** Variation of the effectiveness factor, allometric exponents ($F_k$) with $G_{OD}$ number: Spherical geometry COS-$O_2$ model. With known $F_k$ from experimental data of organ k, determine $G_{OD}$, for each vital organ k as 2.8, 4.2, 11 and 240 for k = K, H, Br and L with corresponding effective factors (=$Y_{O2avg}/Y_{O2cap-IF}$) of 0.85, 0.8, 0.63 and 0.18 (row 7, Table 4). With exponent at −0.25, the BS operates around an average value of $G_{OD}$ = 25 and overall effective factor of 0.48.

**Table 4.** Dimensionless Organ Mass and Allometric Coefficients for Organ Mass and Organ Metabolic Rate. $G_{OD}$ Number for Vital Organs based on COS-$O_2$ Model, Spherical. $F_k$ Values from Table 4 of [20]. mk* for 6 species with body mass ranging from 0.45 kg to 65 kg.

| # | Allometric Constants | Kidneys | Brain | Heart | Liver | Residual |
|---|---|---|---|---|---|---|
| 1 | $g_{k,6} = (c_{k,6}/c_{K,6})$ | 1 | 1.71 | 3.14 | 9.43 | 268.29 |
| 2 | $h_{k,6} = (d_{k,6}-d_{K,6})$ | 0 | 0.13 | −0.09 | 0.02 | 0.16 |
| 3 | $m_{k,6}* = (m_{k,6}/m_{K,6})$ | 1 | 3.3–2.1 | 1.6–3.1 | 9.3–10.3 | - |
| 4 | $F_k$ | −0.094 | −0.184 | −0.122 | −0.31 | - |
| 5 | $G_{OD,k}$ from $F_k$ (6 species) | 2.8 | 11 | 4.2 | 240 | |
| 6 | Median $mk*$ (6 species) with $c_k$ and $d_k$ for 6 species | 1 | 2.66 | 2.28 | 9.82 | 384.91 |
| 7 | Effectiveness factor or $Y_{O2\,avg}/Y_{O2cap-IF}$ | 0.85 | 0.63 | 0.8 | 0.18 | |

Effect of mass of organ-111 species: Wang et al. [51] collected experimental data on the masses of vital organs of 111 species with body mass ranging from 0.0075 (shrew) to 6650 kg (elephant) (see Table A1 in Appendix B) and presented the following curve fit for the product of $\dot{q}_{k,m}$ and $m_k$:

$$\dot{q}_k = \dot{q}_{k,m} \times m_k = \left(e_{k,6}\, m_B^{f_{k,6}}\right) m_k = i_{k,111}\, m_B^{j_{k,111}},\ k = Br, H, K, L \qquad (26)$$

Values for $i_{k,111}$ and $j_{k,111}$ presented by Wang et al. for 111 species were adopted from [51] and are shown in rows 1 and 2 of Table 5. For $\dot{q}_{k,m}$, the authors of ref. [51] use $e_{k,6}$ and $f_{k,6}$ values (based on 6 species; see Tables 2 and 4). By adding ($\dot{q}_k$) over all organ masses of 111 species, they verified Kleiber's law (Equation (11)) and show that

a = 3.21 W/(kg $^{0.754}$), b = 0.754, and b' = −0.246. However, allometric laws for the organ masses based on 111 species were not presented. Since Wang et al. used data on $e_{k,6}$ and $f_{k,6}$ for organ $SMR_k$, one can use the following relation to extract organ mass allometric coefficients $c_{k,111}$ and $d_{k,111}$.

$$\dot{q}_{k,m} \times m_k = \left( e_{k,6}\, m_B{}^{f_{k,6}} \right) \left( c_{k,111}\, m_B{}^{d_{k,111}} \right) = i_{k,111}\, m_B{}^{j_{k,111}}, \quad k = Br, H, K, L \quad (27)$$

The values for $c_{k,111}$ and $d_{k,111}$ are presented in rows 3 and 4 of Table 5 and values for $c_{k,6}$ and $d_{k,6}$ are provided in parentheses in the same table for comparison. See Table A1 in the Appendix for the data table for the masses of organs of 111 species tabulated in ref. [51]. The normalized organ mass relation for 111 species is given as:

$$m_k{}^* = \frac{m_k}{m_K} = \left( \frac{c_{k,111}}{c_{K,111}} \right) m_B{}^{d_{k,111}-d_{K,111}} = g_{k,111} m_B{}^{h111}, \ g_{k,111} = \left( \frac{c_{k,111}}{c_{K,111}} \right), \ h_{k,111} = d_{k,111} - d_{K,111} \quad (28)$$

Note that $h_{k,111}$ (= $d_{k,111} - d_{K,111}$) is extremely low for vital organs indicating a narrow variation in $m_k{}^*$. Hence the variation in $m_k{}^*$ is extremely small when the body mass changed by a factor of 900,000. The $F_k$ values remain almost unchanged (row 5, Table 5). The values $c_{k,111}$ compared to $c_{k,6}$ and $d_{k,111}$ compared to $d_{k,6}$ are not much different.

Effect of Geometry: The results presented for $F_k$ vs. $m_k{}^*$ (Figure 11) are based on spherical geometry. Rather, if the Krogh cylinder model is adopted for the same $S_{cap,m}$ (same capillary surface area to mass ratio) it is equivalent to the same S/V ratio (=$S_{cap,m}{}^*\rho$). As seen in Figure 10, the effectiveness factors are close to each other for these geometries and hence, for $F_k$, or $S_{cap,m}$, the values will not differ much from each other.

**Table 5.** Dimensionless Organ Mass and Allometric Coefficients for Organ Mass and Organ Metabolic Rate. $G_{OD}$ Number for Vital Organs based on COS-$O_2$ Model, Spherical. $F_k$ Values from Table 4 of [20]. mk* for 111 species with body mass ranging from 0.0075 kg to 6650 kg.

| ## | Allometric Constants | Kidneys | Brain | Heart | Liver | Residual |
|---|---|---|---|---|---|---|
| 1 | $i_{k,111}$ | 0.211 | 0.249 | 0.233 | 0.947 | 1.364 |
| 2 | $j_{k,111}$ | 0.7441 | 0.8137 | 0.6446 | 0.6046 | 0.8402 |
| 3 | $c_{k,111} = i_{k,111}/e_{k,6}$ {ck,6} | 0.0063 (0.007) | 0.011 (0.011) | 0.0058 (0.006) | 0.029 (0.033) | 0.94 (0.939) |
| 4 | $d_{k,111} = j_{k,111} - f_{k,6}$ {dk,6} | 0.83 (0.85) | 0.79 (0.76) | 0.93 (0.98) | 0.8723 (0.87) | 1.01 (1.01) |
| 5 | $F_{k,111}$ ($F_{k,6}$) | −0.101 (−0.098) | −0.181 (−0.187) | −0.126 (−0.121) | −0.307 (−0.308) | −0.166 (−0.165) |
| 6 | $g_{k,111} = (c_{k,111}/c_{K,111})$ | 1 | 3.42 | 1.84 | 9.07 | 298.05 |
| 7 | $h_{k,111} = (d_{k,111} - d_{K,111})$ | 0 | −0.041 | 0.10 | 0.045 | 0.18 |
| 8 | $m_{k,111}{}^* = (m_{k,111}/m_{K,111})$ | 1 | 3.5–2.8 | 1.7–2.9 | 8.8–11.2 | - |

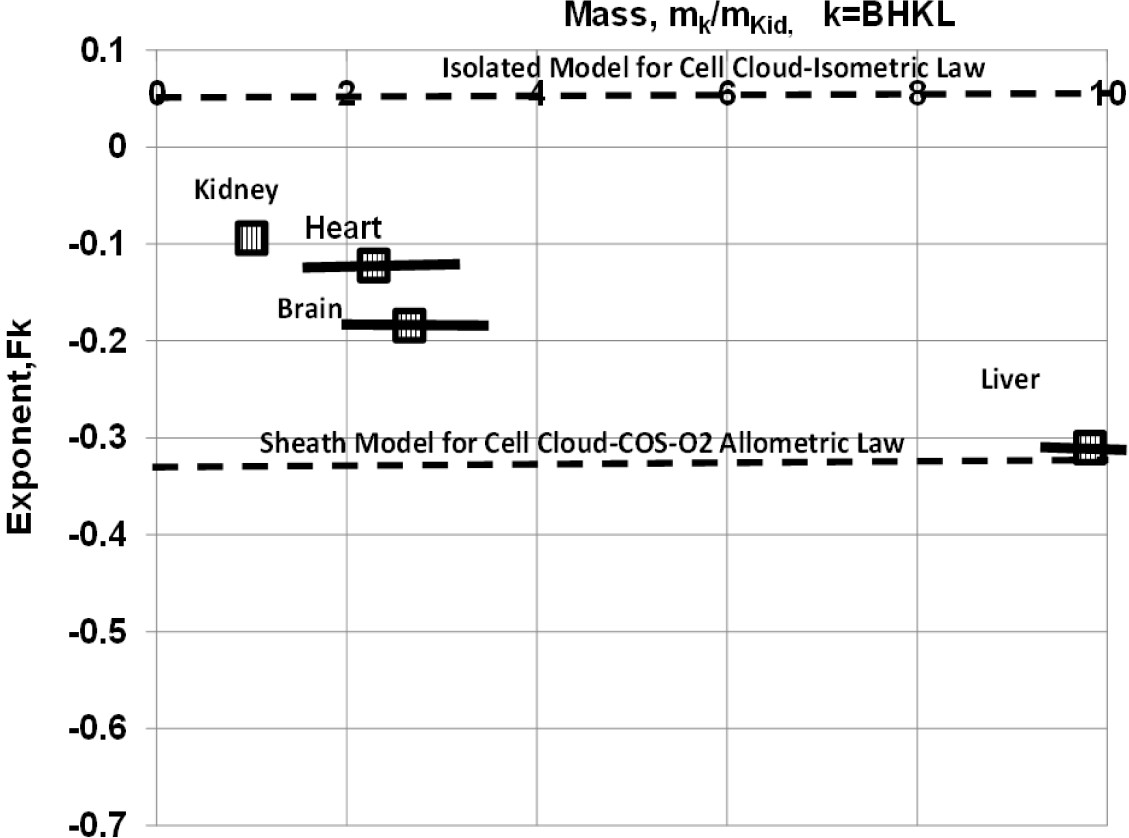

**Figure 12.** Experimental data on allometric exponents vs. the non-dimensional organ mass for the liver, brain, heart, and kidney of 6 species with body mass ranging from 0.45 to 65 kg and comparison with the predicted limits $-\frac{1}{3} < F_k < 0$ for the COS model. $F_k$ values based on experimental data for 111 species with body mass ranging from 0.0075 to 6650 kg are not shown since they are very close to $F_k$ based on 6 species since only $d_k$ values (Table 5) change slightly for 111 species (Table 5) while $f_k$ Equation (7) remains constant.

### 5.6. Whole Body Allometric Law for Metabolic Rate

Various approaches have been undertaken to validate Kleiber's law: surface area rule, oxygen consumption, and capillary surface area rule [52], heterogeneous model of Wang, etc. [17]. (See review in [53]).

If one assumes that all organs within the body are equally larger sized, then Kleiber's law for the whole body will yield $F_k = b' = -1/3$. Similarly, if all organs are extremely small or equally small, then $F_k = 0$ (Figure 11), and Kleiber's law for the whole body will yield $F_k = b' = 0$. However, a BS contains organs of different sizes and hence the exponent $b' = -1/4$ must satisfy the inequality $-1/3 < b' < 0$ if organ sizes vary from smaller to larger. Wang et al. summed up metabolic contribution by five significant organs (Br, H, K, L and R) within the body (Equation (7)) using $f_k$ values based on body mass or $F_k$ values based on organ mass, obtained Kleiber's law with $b' = -1/4$, which satisfies the above inequality [17]. The relation between $F_k$ and organ mass $m_k$ leads to Kleiber's law for the whole body. It appears that the" Kleiber's Law or the 3/4 Rule may neither be a law nor a Rule" [53] but appears to be consequence of combined contributions from all organs of various sizes with different effectiveness factors or oxidation at different concentrations of $O_2$.

### 5.7. $G_{OD}$ Number Estimation of Vital Organs

The definition of $G_{OD,k}$ can be used to estimate the $G_{OD,k}$, the number for vital organs, if the constant of proportionality, $C_{Ch,k}$, is known for each organ; the constants depend on Michaelis Menten (MM) constants $k_{MM}$ in kinetics expression (Equation (16))

$$\dot{w}_{O2,cell} = \dot{w}_{O2,cell,max} \left( \frac{Y_{O2}}{Y_{O2} + k_{MM}} \right) \qquad (29)$$

for each organ if kinetics control and effective diffusivity if diffusion control. On the other hand, if the experimental data for $F_k$ is known, the corresponding $G_{OD, k}$ can be determined using Figure 11. Since $F_k$ is close to the lower limit of $-0.333$, the liver, the largest vital organ, has the highest $G_{OD}$ number (=Thiele Modulus$^2$), while the kidney has the lowest values. The same figure shows the allometric exponent for the whole body (Figure 11) and hence, for the six species considered (0.48–65 kg body mass), the BS operates with an overall $G_{OD}$ of 26 and effectiveness factor of 0.48 (i.e., oxidation for whole body is at 48% capillary oxygen concentration) for spherical geometry which is close to $\psi_T^2 = 25$ (presented in the Appendix of Ref. [54]). Note that [54] uses the whole-body approach and the Thiele modulus in modeling the metabolism. While Thiele modulus is typically used when cells in cell cloud consume oxygen under first order kinetics control, the $G_{OD}$ as defined here is more general and can be used for kinetics control (for which $G\# = \Psi_T^2$, ref. [26,29]), diffusion, or combined control.

### 5.8. Incipient ODC or Group Combustion and Incipient ODM

The incipient combustion number, $G_{INC}$, is transferred from the combustion literature to incipient number, $G_{OD, INC}$, for organs where oxygen concentration reaches "extinction" levels at the core of cell clouds resulting in cessation of oxidation and the inception of fermentation, a precursor to the creation of cancer cells. Cancer cells rely more on glycolysis (which produces ingredients for growth of new cells) due to mitochondrial impairment and oxygen deficiency [55].

As the number of cells per unit volume (n) is increased or when the enzyme activity is enhanced, there is an increasing amount of consumption of oxygen by cells closer to capillaries which lowers the oxygen concentration at the core (r = 0). At $G_{OD, Inc}$, the $Y_{O2}$ at r = 0 reaches the lethal mass fraction $Y_{O2, lethal}$ (about 1 ppm) below which the cells in the core undergo only glycolysis. This condition for cell clouds is like the extinction condition in combustion literature and $G_{INC}$ at this condition is termed as the incipient group combustion in combustion literature. Table 3 row 12 presents the relation for $G_{OD, Inc}$. At $G_{OD, INC}$, the cells at the core become oxygen deficient and the oxidation reaction ceases. For example, if $Y_{O2leth}/Y_{O2cap-IF}$ is set at 0.2, then $G_{OD, INC} = 10$. Note that oxygen deficiency is one of the causes for creation of cancer cells but not necessarily the cause. Tumor cells seem to adopt glycolysis as a metabolic pathway for energy conversion independent of oxygen levels within the cells. Thus, one may chronologically monitor the $G_{OD}\#$ of each organ and if it keeps increasing over a period, there is a likelihood of the formation of cancer cells.

According to the present hypothesis, a larger species is more likely to develop cancer, although there is some existing evidence to the contrary [56]. On the other hand, a large organ like the liver can afford impairment, while small organs like the pancreas cannot and they could be compromised with only a few tumor cells [57]. Further, the present hypothesis does not consider the fact that a reduced metabolic rate reduces the number of electrons transferred which reduces the oxygen radical concentration resulting in a reduced oncogenic mutation and hence a reduced rate of buildup of cancer cells [56]. Thus, the competitive effects seem to be present in each generation of cancer cells. Typically, the highest rate of cancer occurs in the breast followed by lung cancer and prostate. The future work should track $G_{OD,k}$ for these organs for any sign of gradual increase in $G_{OD,k}$. Similarly, in COVID-19 patients with 50% saturation in arterial blood, the location of $Y_{O2leth}$

is close to the capillaries on the surface, leaving a large anaerobic core or lethal volume with reduced ATP production and resulting in the loss of vital functions.

### 6. Conclusions

1.  The dimensionless number ($G$) used in combustion science was modified to $G_{OD}$ for the application to OD metabolisms in organs.
2.  The rationale for negative exponents ($F_k$) in the allometric law for SMR of vital organs, $\dot{q}_{k,m} \propto m_k^{F_k}$, is provided through the adoption of group combustion or OD theory from combustion science within biology. The exponent $F_k$ satisfies the inequality $-\frac{1}{3} < F_k < 0$ for COS-$O_2$ model. Thus, the allometric law for an organ is shown to be an empirical approximation of effectiveness chart.
3.  The negative exponent $b'$ in the allometric law for the SMR of the whole body satisfies the inequality $-\frac{1}{3} < b' < 0$ with $b' = -1/4$ (Kleiber's law). The negative exponent $b' = -1/4$ is due to distributed sizes of various organs with differing values for $F_k$.
4.  Based on experimental data from $F_k$, the $G_{OD}$ numbers were estimated for vital organs as 2.8, 4.2, 9, and 240 for the kidney, heart, brain, and liver, respectively. The average $G_{OD}$ for the whole body of all BS with an equivalent organ is about 26 with effectiveness factor of 0.48 (i.e., oxidation of all tissues at 48% capillary oxygen concentration), considering spherical geometries for cell clouds.
5.  While the hypoxic condition of the whole body is characterized by the saturation percentage, the introduction of the $G_{OD}$ number characterizes OD at organ level.
6.  Glycolysis is inhibited when there is a high level of ATP within cells and vice versa; thus, reduced $O_2$ concentration results in a reduced ERR per unit volume which results in reduced ATP concentration, which promotes glycolysis pathway [58,59]. Cancer and virus cells, including those of COVID-19 patients, rely on the glycolysis pathway to provide the building blocks for uncontrollable cell growth.

**Funding:** The seeds for the research on group/oxygen-deficient combustion of carbon clouds were planted by the earlier funding from DOE-Pittsburgh: DE-FG22-90 PC 90310. DE-FG 22-88 PC 88937, DE-FG 22-85 PC 80528 and Department of Energy—Morgantown: DOE-METC DE-AC21-86 MC 23256. The current research was initiated after observing the similarity of the results on energy release rate from carbon clouds with those from organs (cell clouds). The author wishes to acknowledge the partial support of the research funds from Paul Pepper Professorship.

**Institutional Review Board Statement:** Not applicable.

**Informed Consent Statement:** Not applicable.

**Data Availability Statement:** Not applicable.

**Acknowledgments:** Vishal M. Gohil, Biochemistry and Biophysics, Texas A&M University, for fruitful discussions on mitochondria and metabolism. Mathew Miller, for help in generating Hb-$O_2$ equilibrium curves and Figures 5 and 7. Megan Simpson of the Department of Mechanical Engineering, Texas A&M University, for English editing of manuscript.

**Conflicts of Interest:** The author declares no conflict of interest.

### Abbreviations/Nomenclature

**Nomenclature**

| | |
|---|---|
| $A_k$ | see Equation (22), pre-exponential factor |
| $a, b, b'$ | allometric coefficients for metabolism |
| $C_{ch,\ cell}$ | Characteristic Oxygen Consumption Rate for cell, g/s, {Table 3} |
| $C_{ch,\ p}$ | Characteristic Oxygen Consumption Rate for carbon particle, g/s, {Table 3} |
| ck | pre-exponent in the allometry relation for mass of organ |
| D | Diffusivity, cm$^2$/s |
| $D_{eff}$ | Effective Diffusion Coefficient, cm$^2$/s |

| | |
|---|---|
| $d_{cell}$ | Diameter of cell, cm |
| $d_k$ | exponent in allometry relation for mass of organ |
| $d_p$ | Diameter of particle, cm |
| D | Diffusion Coefficient of Molecular Oxygen |
| $D_{eff}$ | Effective diffusion Coefficient |
| $e_k$ | pre-exponent in the allometry relation for metabolic rate |
| $F_k$ | Allometric exponent, organ mass based $\{\dot{q}_{k,m} = E_k\, m_k^{F_k}\}$ |
| $f_k$ | Allometric exponent, body mass based allometry used in $\dot{q}_{k,m} = e_k\, m_B^{f_k}\}$ |
| G | Group combustion number in combustion science {Table 3} |
| $G_{OD}$ | Group metabolism number in biology to indicate extent of Oxygen Deficiency (OD) |
| $G_{OD,\,kin}$ | Group metabolism number for organ k under kinetics control $\{= \Psi_T^2$ or Thiele Modulus$^2$, $\{\propto R_k^2 \propto m_k^{2/3}\}$ |
| $G_{k,\,OD,\,diff}$ | Group metabolism number for organ k under diffusion control, $\{\propto R_k^2 \propto m_k^{2/3}\}$ |
| $g_k, h_k$ | see Equation (25) |
| (Hb) | Hemoglobin concentration on mass basis, g per DL of blood |
| [Hb] | Hemoglobin concentration on mole basis, moles per DL of blood |
| $H_{O2}$ | Henry's constant for solubility of $O_2$ in blood |
| HV | Heating Value of fuel or nutrient, J/g fuel |
| $HV_{O2}$ | Heating Value per unit mass of $O_2$ consumed, J/g $O_2$ |
| $J_k$ | Pre-exponent in allometry relation for OEF |
| $k_{MM}$ | Constant in MM kinetics, Equation (29), = $Y_{O2}$ at half of maximum metabolic rate |
| $K_n'$ | Equilibrium Constant, (1/mm Hg when n = 1, 1/mm Hg$^2$ n = 2, etc.), n = 1, 2, 3, 4 |
| $L_{MM}$ | Constant in LM kinetics, $Y_{O2}$ at half of maximum oxygen consumption rate $\dot{w}_{O2,p,max}$ |
| $m_B$ | Body mass, kg |
| $m_{cl}$ | cloud mass, kg |
| mk, | k th organ mass, kg, Brain, Heart, Kidney, liver etc. |
| mk* | Non-dimensional mass, = {mass of organ k/Mass of kidney} |
| n | number of particle or cells per unit volume, particles/cm$^3$, or cells/cm$^3$ |
| $p_{O2}$ | Partial pressure of $O_2$, mm of Hg |
| $\dot{Q}$ | Energy released in the form of heat, W |
| $\dot{q}$ | Energy release rate (ERR) or metabolic rate, W |
| $\dot{q}k,\,m$ | Specific Energy Release Rate or SMR (SERR), W/kg of k-th organ |
| $\dot{q}_m$ | Specific Energy Release Rate or SMR (SERR), W/kg |
| r | radius |
| $r_{cap}$ | Capillary radius, cm |
| $R, R_{cl}$ | Cloud radius, cm |
| RQ | Respiratory Quotient, =$CO_2$ moles/$O_2$ moles, Table 1 |
| $S, S_{cap}$ | Capillary surface area, cm$^2$ |
| $S_{cap,\,m}$ | Specific capillary surface area based on mass, cm$^2$/g of tissue, $\{=S/(\rho V)\}$ |
| $s_k$ | See equation (22), exponent |
| V | Volume, = cm$^3$ |
| $\dot{W},\ \dot{W}_{ATP}$ | Energy Released in the form of Work or ATP, W |
| $\dot{w}_{O2},\ \dot{w}_{O2}'',\ \dot{w}_{O2}'''$ | Oxygen Consumption rate, g/s; per unit area, g/(cm$^2$ s), per unit volume, g/(cm$^3$ s) |
| $Y_{O2}$ | Mass fraction of oxygen, (g of $O_2$/g of mixture) |
| δ | Aerobic shell thickness |
| **Greek Symbols** | |
| $\xi$ | Non-dimensional radius, r/R |
| $\rho_{bl}$ | Density of blood, g/cm$^3$ |
| $\psi_T$ | Thiele Modulus, $\sqrt{G_{kin}}$ or $\sqrt{G_{OD,kin}}$ |
| $\eta_{eff}$ | Effectiveness factor based on capillary-IF interface $O_2$ concentration, = $Y_{O2,\,avg}/Y_{O2,\,cap\text{-}IF}$ |
| $\eta_M$ | metabolic efficiency |
| $\nu_{O2}$ | stoichiometric Oxygen |

**Subscripts**

| | |
|---|---|
| A | Alveolar |
| avg | Average |
| a | Arterial |
| B | Body |
| Bl | Blood |
| cap | capillary |
| Cap-IF | interface between capillary and Interstitial Fluid (IF) |
| cl | Cloud |
| eff | effective |
| ext | extinction |
| IF | Interstitial fluid |
| k | organ k, Brain (Br), Heart (H), Kidney (K), liver (L) etc. |
| Leth | Lethal |
| m | per unit mass |
| $O_2$ | oxygen |

**Abbreviations**

| | |
|---|---|
| ATP | Adenosine triphosphate |
| BMA | Body mass based allometry |
| Br, H, K, L, R | Brain, Heart, Kidney, Liver, Residual |
| BS | Biological systems |
| CH | Carbo-hydrate, e.g., glucose |
| COA | Capillary on Axis of Cylinder |
| COS | Capillary on Surface of Cylinder or solid cylinder model |
| ER | Equivalence radio in engineering, $O_2$ used/$O_2$ supplied, ER < 1 dilute mix |
| ERR | Energy Release Rate, W |
| F | Fat |
| GC | Group Combustion |
| ICD | Inter-capillary Distance, cm |
| IF | Interstitial Fluid |
| Iso | isolated |
| LM | Langmuir kinetics |
| MITO | Mitochondria |
| MM | Michaelis Menton kinetics |
| MR | Metabolic rate, W |
| OD | Oxygen Deficiency, oxygen deficient |
| ODC | Oxygen Deficient Combustion |
| ODM | Oxygen deficient metabolism |
| OEF | Oxygen Extraction Fraction (g $O_2$ extracted for metabolism er g O in blood = Equivalence Ratio in Combustion. Science for dilute Combustible Mix) |
| OHb | Oxy-Hemoglobin |
| OMA | Organ mass based allometry |
| OXPHOS | Oxidative phosphorylation |
| P | Protein |
| ppm | parts per million {(g per/g blood) * $10^6$} |
| RBC | Red Blood Cells |
| Sa | Saturation |
| SERR | Specific Energy Release Rate, W/g |
| $SERR_M$ | Specific Energy Release Rate, W/g body mass |
| SMRk | Specific Metabolic Rate (term in Biology for SERR) of organ k, W/g |
| $SMR_M$ | Specific Metabolic Rate of whole body, W/g body mass |

## Appendix A. Solutions for Oxygen Concentrations with Lethal Volume

Consider

$$\frac{d^2Y_{O2}}{d\xi^2} + \left(\frac{2}{\xi}\right)\frac{dY_{O2}}{d\xi} = Y_{O2}\, G_{OD} \tag{A1}$$

Using the boundary conditions $Y_{O2} = Y_{O2, ccap-IF}$ at $\xi = 1$ and $dY_{O2}/d\xi = 0$ at $\xi = \xi_{leth}$. The solution for $Y_{O2}$ with lethal volume is given as

$$\frac{Y_{O2}(\xi)}{Y_{O2cap-IF}} = \frac{\left[\frac{Sinh\left(\sqrt{G_{OD}}\xi\right)}{\sqrt{G_{OD}}\xi} - \frac{\left\{\sqrt{G_{OD}}\,\xi_{leth}-\tanh\left(\sqrt{G_{OD}}\,\xi_{leth}\right)\right\}}{\left\{\sqrt{G_{OD}}\,\tanh\left(\sqrt{G_{OD}}\,\xi_{leth}\right)-1\right\}}\frac{\cosh\left(\sqrt{G_{OD}}\,\xi\right)}{\sqrt{G_{OD}}\xi}\right]}{\left\{\frac{Sinh\left(\sqrt{G_{OD}}\right)}{\sqrt{G_{OD}}} - \frac{\left\{\sqrt{G_{OD}}\,\xi_{leth}-\tanh\left(\sqrt{G_{OD}}\,\xi_{leth}\right)\right\}}{\left\{\sqrt{G_{OD}}\,\xi_{leth}\tanh\left(\sqrt{G_{OD}}\xi_{leth}\right)-1\right\}}\frac{\cosh\left(\sqrt{G_{OD}}\right)}{\sqrt{G_{OD}}}\right\}}, COS\text{-}O_2, \text{Lethal} \tag{A2}$$

where $\xi_{leth}$ can be solved by setting $Y_{O2} = Y_{O2\ leth}$ at $\xi = \xi_{leth}$

$$\frac{Y_{O2,leth}}{Y_{O2cap-IF}} = \frac{\left[\frac{Sinh\left(\sqrt{G_{OD}}\xi_{leth}\right)}{\sqrt{G_{OD}}\xi_{leth}} - \frac{-\left\{\sqrt{G_{OD}}\,\xi_{leth}-\tanh\left(\sqrt{G_{OD}}\,\xi_{leth}\right)\right\}}{\left\{\sqrt{G_{OD}}\,\tanh\left(\sqrt{G_{OD}}\,\xi_{leth}\right)-1\right\}}\frac{\cosh\left(\sqrt{G_{OD}}\,\xi_{leth}\right)}{\sqrt{G_{OD}}\xi_{leth}}\right]}{\left\{-\frac{\left\{\sqrt{G_{OD}}\,\xi_{leth}-\tanh\left(\sqrt{G_{OD}}\,\xi_{leth}\right)\right\}}{\left\{\sqrt{G_{OD}}\,\xi_{leth}\tanh\left(\sqrt{G_{OD}}\xi_{leth}\right)-1\right\}}\frac{\cosh\left(\sqrt{G_{OD}}\right)}{\sqrt{G_{OD}}} + \frac{Sinh\left(\sqrt{G_{OD}}\right)}{\sqrt{G_{OD}}}\right\}}, COS\text{-}O_2, \text{Lethal} \tag{A3}$$

By letting $\xi_{leth} = 0$, (i.e., solution with no lethal volume), Equation (A2) becomes

$$\frac{Y_{O2}(\xi)}{Y_{O2cap-IF}} = \frac{1}{\xi}\left[\frac{Sinh\left(\sqrt{G_{OD}}\xi\right)}{Sinh\left(\sqrt{G_{OD}}\right)}\right], COS\text{-}O_2, \text{No Lethal vol} \tag{A4}$$

which is the same as the solution presented in row 10 of Table 3.

## Appendix B. Collected Organ Mass and Metabolism Data On 111 Species

**Table A1.** Data on organ mass in kg, normalized mass and specific metabolic rates of vital organs and computed overall metabolic rate of whole body for 111 species. Mass and specific metabolic rate data for organs and whole body adopted from [51] and specific metabolic rate of organs (SMR) converted into Watts. Normalized organ mass $\{m_k^* = m_k/m_K\}$ computed in current manuscript. $q_{k,m}$ (SMR) W/kg, k = L, Br, H, K and R; $q_{Het}$: Metabolic rate of whole body-Heterogeneous approach, W; $q_{Het,m}$: Specific Metabolic rate of whole body-Heterogeneous approach, W/kg. $m_B = m_L + m_{Br} + m_H + m_K + m_R$.

| Species | M | $q_{L,m}$ | $q_{Br,m}$ | $q_{H,m}$ | $q_{K,m}$ | $q_{R,m}$ | $m_L$ | $m_{Br}$ | $m_H$ | $m_K$ | $m_R$ | $q_{Het}$ | $q_{Het,m}$ | $m_L$ * | $m_{Br}$ * | $m_H$ * |
|---|---|---|---|---|---|---|---|---|---|---|---|---|---|---|---|---|
| Sorex araneus | 0.0075 | 123 | 43 | 77 | 50 | 3 | 0.00038 | 0.00015 | 0.00011 | 0.00011 | 0.0068 | 0.09 | 11.86 | 6.91 | 2.73 | 2.00 |
| Crocidura russula | 0.0100 | 115 | 42 | 75 | 49 | 3 | 0.00055 | 0.00017 | 0.00008 | 0.00013 | 0.0086 | 0.10 | 10.85 | 8.46 | 2.62 | 1.23 |
| Lasiurus borealis | 0.0140 | 105 | 40 | 72 | 48 | 3 | 0.00035 | 0.00017 | 0.00014 | 0.00011 | 0.013 | 0.10 | 7.17 | 6.36 | 3.09 | 2.55 |
| Lasionycteris noctivagans | 0.0150 | 102 | 39 | 71 | 47 | 3 | 0.00033 | 0.00016 | 0.00016 | 0.00013 | 0.014 | 0.10 | 6.68 | 5.08 | 2.46 | 2.46 |
| Mus musculus | 0.0150 | 102 | 39 | 71 | 47 | 3 | 0.00068 | 0.00036 | 0.00007 | 0.00028 | 0.014 | 0.14 | 9.49 | 4.86 | 2.57 | 0.50 |
| Myodes glareolus | 0.0150 | 101 | 39 | 71 | 47 | 3 | 0.00067 | 0.00035 | 0.0001 | 0.00024 | 0.014 | 0.14 | 9.10 | 5.58 | 2.92 | 0.83 |
| Microtus agrestis | 0.0150 | 101 | 39 | 71 | 47 | 3 | 0.00063 | 0.00039 | 0.00012 | 0.00017 | 0.014 | 0.14 | 8.86 | 7.41 | 4.59 | 1.41 |
| Neomys fodiens | 0.0160 | 101 | 39 | 70 | 47 | 3 | 0.00055 | 0.00025 | 0.00014 | 0.00022 | 0.015 | 0.13 | 8.14 | 5.00 | 2.27 | 1.27 |
| Blarina brevicauda | 0.0180 | 97 | 38 | 69 | 47 | 3 | 0.00093 | 0.00032 | 0.00018 | 0.00021 | 0.016 | 0.17 | 9.64 | 8.86 | 3.05 | 1.71 |
| Apodemus sylvaticus | 0.0180 | 97 | 38 | 69 | 47 | 3 | 0.0011 | 0.00057 | 0.00014 | 0.00026 | 0.016 | 0.19 | 10.70 | 8.46 | 4.38 | 1.08 |
| Microtus | 0.0210 | 93 | 37 | 68 | 46 | 3 | 0.0011 | 0.00058 | 0.00015 | 0.00036 | 0.019 | 0.19 | 8.91 | 6.11 | 3.22 | 0.83 |
| Peromyscus leucopus | 0.0220 | 92 | 37 | 68 | 46 | 3 | 0.0012 | 0.00074 | 0.00015 | 0.0003 | 0.02 | 0.22 | 9.88 | 8.00 | 4.93 | 1.00 |
| Apodemus flavicollis | 0.0250 | 89 | 37 | 67 | 45 | 3 | 0.001 | 0.00061 | 0.00018 | 0.00034 | 0.023 | 0.20 | 7.94 | 5.88 | 3.59 | 1.06 |
| Nyctalus noctula | 0.0260 | 88 | 36 | 66 | 45 | 3 | 0.0005 | 0.00032 | 0.00037 | 0.00013 | 0.024 | 0.15 | 5.76 | 7.69 | 4.92 | 5.69 |
| Microtus arvalis | 0.0270 | 87 | 36 | 66 | 45 | 3 | 0.0019 | 0.00039 | 0.00019 | 0.00055 | 0.024 | 0.28 | 10.31 | 6.91 | 1.42 | 0.69 |
| Mouse | 0.0280 | 86 | 36 | 66 | 45 | 3 | 0.0018 | 0.0005 | 0.00016 | 0.00051 | 0.025 | 0.27 | 9.64 | 7.06 | 1.96 | 0.63 |
| Gerbillus perpallidus | 0.0300 | 85 | 36 | 65 | 45 | 3 | 0.001 | 0.00058 | 0.00013 | 0.00027 | 0.028 | 0.19 | 6.54 | 7.41 | 4.30 | 0.96 |
| Mustela nivalis | 0.0320 | 83 | 35 | 65 | 44 | 3 | 0.0016 | 0.0018 | 0.00036 | 0.00043 | 0.028 | 0.31 | 9.49 | 7.44 | 8.37 | 1.67 |
| Acomys minous | 0.0420 | 77 | 34 | 63 | 43 | 2 | 0.0009 | 0.0009 | 0.00018 | 0.00032 | 0.04 | 0.23 | 5.33 | 5.63 | 5.63 | 1.13 |
| Jaculus jaculus | 0.0480 | 75 | 33 | 62 | 43 | 2 | 0.0011 | 0.0012 | 0.00045 | 0.00029 | 0.045 | 0.27 | 5.67 | 7.59 | 8.28 | 3.10 |
| Rhabdomys pumilio | 0.0500 | 74 | 33 | 61 | 43 | 2 | 0.0018 | 0.0006 | 0.00021 | 0.00041 | 0.047 | 5.96 | 0.00 | 8.78 | 2.93 | 1.02 |
| Talpa europaea | 0.0510 | 73 | 33 | 61 | 43 | 2 | 0.0015 | 0.001 | 0.00031 | 0.00036 | 0.048 | 0.30 | 5.71 | 8.33 | 5.56 | 1.72 |
| Glaucomys volans | 0.0550 | 72 | 33 | 61 | 43 | 2 | 0.0029 | 0.0019 | 0.00056 | 0.00059 | 0.049 | 0.45 | 8.09 | 9.83 | 6.44 | 1.90 |
| Arvicola terrestris | 0.0620 | 70 | 32 | 60 | 42 | 2 | 0.0026 | 0.0011 | 0.00028 | 0.0007 | 0.057 | 0.39 | 6.34 | 7.43 | 3.14 | 0.80 |
| Glis glis | 0.0830 | 64 | 31 | 58 | 41 | 2 | 0.0032 | 0.0015 | 0.00048 | 0.00068 | 0.078 | 0.48 | 5.71 | 9.41 | 4.41 | 1.41 |
| Tamias striatus | 0.1040 | 61 | 30 | 56 | 40 | 2 | 0.0029 | 0.0024 | 0.00066 | 0.00081 | 0.097 | 5.04 | 0.00 | 7.16 | 5.93 | 1.63 |

**Table A1.** *Cont.*

| Species | M | q$_{L,m}$ | q$_{Br,m}$ | q$_{H,m}$ | q$_{K,m}$ | q$_{R,m}$ | m$_L$ | m$_{Br}$ | m$_H$ | m$_K$ | m$_R$ | q$_{Het,l}$ | q$_{Het,m}$ | m$_L$ * | m$_{Br}$ * | m$_H$ * |
|---|---|---|---|---|---|---|---|---|---|---|---|---|---|---|---|---|
| Octodon degus | 0.1290 | 57 | 29 | 55 | 40 | 2 | 0.0048 | 0.0019 | 0.00041 | 0.0011 | 0.121 | 4.99 | 0.00 | 8.73 | 3.45 | 0.75 |
| Tupaia glis | 0.1410 | 56 | 29 | 54 | 39 | 2 | 0.0034 | 0.0034 | 0.00117 | 0.0011 | 0.132 | 4.69 | 0.00 | 6.18 | 6.18 | 2.13 |
| Rat | 0.1500 | 55 | 28 | 54 | 39 | 2 | 0.0092 | 0.0023 | 0.0007 | 0.0014 | 0.136 | 6.25 | 0.00 | 13.1 | 3.29 | 1.00 |
| Cebuella | 0.1630 | 54 | 28 | 53 | 39 | 2 | 0.0135 | 0.0044 | 0.00086 | 0.0019 | 0.142 | 1.25 | 0.00 | 14.2 | 4.63 | 0.91 |
| Rattus norvegicus | 0.2100 | 50 | 27 | 52 | 38 | 2 | 0.0092 | 0.0023 | 0.00087 | 0.0015 | 0.196 | 4.75 | 0.00 | 12.3 | 3.07 | 1.16 |
| Cheirogaleus medius | 0.2310 | 49 | 27 | 51 | 38 | 2 | 0.0063 | 0.0028 | 0.00093 | 0.001 | 0.22 | 3.80 | 0.00 | 12.6 | 5.60 | 1.86 |
| Rat | 0.2500 | 48 | 26 | 51 | 37 | 2 | 0.012 | 0.002 | 0.00094 | 0.0021 | 0.233 | 4.73 | 0.00 | 11.4 | 1.90 | 0.90 |
| Mustela erminea | 0.2590 | 48 | 26 | 51 | 37 | 2 | 0.01 | 0.0057 | 0.0025 | 0.0023 | 0.238 | 4.89 | 0.00 | 8.70 | 4.96 | 2.17 |
| Helogale parvula | 0.2600 | 48 | 26 | 51 | 37 | 2 | 0.0111 | 0.0052 | 0.0015 | 0.0025 | 0.24 | 4.84 | 0.00 | 8.88 | 4.16 | 1.20 |
| Sciurus vulgaris | 0.2750 | 47 | 26 | 50 | 37 | 2 | 0.0055 | 0.0063 | 0.0017 | 0.0017 | 0.259 | 3.77 | 0.00 | 6.47 | 7.41 | 2.00 |
| Callithrix jacchus | 0.3120 | 45 | 26 | 49 | 37 | 2 | 0.0178 | 0.0073 | 0.0028 | 0.0029 | 0.281 | 5.57 | 0.00 | 12.3 | 5.03 | 1.93 |
| Saguinus fuscicollis | 0.3300 | 45 | 25 | 49 | 37 | 2 | 0.0144 | 0.0078 | 0.0033 | 0.0019 | 0.303 | 1.60 | 0.00 | 15.2 | 8.21 | 3.47 |
| Rat | 0.3370 | 44 | 25 | 49 | 37 | 2 | 0.008 | 0.0019 | 0.001 | 0.0023 | 0.324 | 1.10 | 0.00 | 6.96 | 1.65 | 0.87 |
| Rat (Wistar) | 0.3900 | 43 | 25 | 48 | 36 | 2 | 0.0143 | 0.0019 | 0.0011 | 0.0028 | 0.37 | 1.44 | 0.00 | 10.2 | 1.36 | 0.79 |
| Sciurus niger | 0.4120 | 42 | 25 | 48 | 36 | 2 | 0.0107 | 0.0075 | 0.0025 | 0.003 | 0.389 | 1.52 | 0.00 | 7.13 | 5.00 | 1.67 |
| Sciurus carolinensis | 0.5960 | 38 | 23 | 46 | 35 | 2 | 0.0164 | 0.0075 | 0.0028 | 0.0032 | 0.566 | 1.94 | 0.00 | 10.3 | 4.69 | 1.75 |
| Saguinus oedipus | 0.6240 | 38 | 23 | 46 | 35 | 2 | 0.0209 | 0.01 | 0.0037 | 0.0031 | 0.586 | 2.21 | 0.00 | 13.5 | 6.45 | 2.39 |
| Mustela putorius | 0.6400 | 37 | 23 | 45 | 35 | 2 | 0.0288 | 0.0104 | 0.0048 | 0.004 | 0.592 | 2.60 | 4.06 | 14.4 | 5.20 | 2.40 |
| Leontopithecus chrysomelas | 0.6420 | 37 | 23 | 45 | 35 | 2 | 0.0189 | 0.0132 | 0.0038 | 0.0041 | 0.602 | 2.27 | 3.54 | 9.22 | 6.44 | 1.85 |
| Guinea pig | 0.8000 | 35 | 22 | 44 | 34 | 2 | 0.027 | 0.0047 | 0.0023 | 0.0056 | 0.76 | 2.49 | 3.11 | 9.64 | 1.68 | 0.82 |
| Potorous tridactylu | 0.8090 | 35 | 22 | 44 | 34 | 2 | 0.0237 | 0.0114 | 0.0048 | 0.0062 | 0.763 | 2.65 | 3.28 | 7.65 | 3.68 | 1.55 |
| Erinaceus europaeus | 0.9500 | 34 | 22 | 43 | 34 | 1 | 0.0496 | 0.0043 | 0.0055 | 0.0089 | 0.881 | 3.59 | 3.78 | 11.2 | 0.97 | 1.24 |
| Sylvilagus floridanus | 0.9720 | 33 | 22 | 43 | 33 | 1 | 0.032 | 0.0079 | 0.0048 | 0.0063 | 0.921 | 3.00 | 3.08 | 10.2 | 2.51 | 1.52 |
| Ondatra zibethicus | 0.9910 | 33 | 22 | 43 | 33 | 1 | 0.026 | 0.0047 | 0.003 | 0.0058 | 0.952 | 2.67 | 2.69 | 8.97 | 1.62 | 1.03 |
| Saimiri boliviensis | 1.0000 | 33 | 22 | 43 | 33 | 1 | 0.0194 | 0.029 | 0.0065 | 0.0067 | 0.941 | 3.14 | 3.13 | 5.79 | 8.66 | 1.94 |
| Martes foina | 1.4100 | 30 | 21 | 41 | 32 | 1 | 0.0349 | 0.019 | 0.0098 | 0.0073 | 1.335 | 3.92 | 2.79 | 9.56 | 5.21 | 2.68 |
| Mephitis | 1.4500 | 30 | 21 | 41 | 32 | 1 | 0.0174 | 0.0098 | 0.006 | 0.0066 | 1.409 | 3.11 | 2.15 | 5.27 | 2.97 | 1.82 |
| Trichosurus vulpecula | 1.5500 | 29 | 20 | 41 | 32 | 1 | 0.0332 | 0.0127 | 0.009 | 0.0135 | 1.482 | 4.03 | 2.60 | 4.92 | 1.88 | 1.33 |
| Martes martes | 1.6000 | 29 | 20 | 41 | 32 | 1 | 0.0379 | 0.0205 | 0.0108 | 0.0088 | 1.525 | 4.29 | 2.68 | 8.61 | 4.66 | 2.45 |
| Cebus apella | 1.7500 | 29 | 20 | 40 | 32 | 1 | 0.0493 | 0.0508 | 0.0134 | 0.0104 | 1.626 | 5.42 | 3.11 | 9.48 | 9.77 | 2.58 |
| Eulemur macaco macaco | 1.8800 | 28 | 20 | 40 | 32 | 1 | 0.0778 | 0.0242 | 0.0091 | 0.0142 | 1.75 | 5.76 | 3.07 | 10.9 | 3.41 | 1.28 |
| Chrotagale owstoni | 1.9600 | 28 | 20 | 40 | 32 | 1 | 0.0441 | 0.0233 | 0.0116 | 0.0128 | 1.868 | 4.99 | 2.53 | 6.89 | 3.64 | 1.81 |
| Vulpes corsac | 2.0800 | 27 | 20 | 40 | 31 | 1 | 0.0356 | 0.0341 | 0.0217 | 0.0088 | 1.975 | 5.33 | 2.56 | 8.09 | 7.75 | 4.93 |
| Lemur catta | 2.0800 | 27 | 20 | 40 | 31 | 1 | 0.0729 | 0.0228 | 0.0117 | 0.0112 | 1.956 | 5.76 | 2.77 | 13.0 | 4.07 | 2.09 |
| Eulemur fulvus fulvus | 2.5000 | 26 | 19 | 39 | 31 | 1 | 0.0434 | 0.0225 | 0.0118 | 0.0095 | 2.413 | 5.33 | 2.12 | 9.14 | 4.74 | 2.48 |
| Felis silvestris | 2.5700 | 26 | 19 | 39 | 31 | 1 | 0.0502 | 0.0381 | 0.0103 | 0.0154 | 2.459 | 5.91 | 2.31 | 6.52 | 4.95 | 1.34 |
| Didelphis virginiana | 2.6300 | 26 | 19 | 38 | 31 | 1 | 0.1573 | 0.0083 | 0.0121 | 0.0229 | 2.433 | 8.33 | 3.17 | 13.7 | 0.72 | 1.06 |
| Aonyx cinerea | 2.6800 | 25 | 19 | 38 | 31 | 1 | 0.1064 | 0.0359 | 0.0151 | 0.0306 | 2.487 | 7.99 | 2.98 | 6.95 | 2.35 | 0.99 |
| Leopardus geoffroyi | 3.1000 | 24 | 18 | 38 | 30 | 1 | 0.0584 | 0.0321 | 0.016 | 0.0307 | 2.963 | 7.12 | 2.30 | 3.80 | 2.09 | 1.04 |
| Lepus europaeus | 3.3400 | 24 | 18 | 37 | 30 | 1 | 0.0904 | 0.0148 | 0.0289 | 0.0185 | 3.186 | 7.85 | 2.35 | 9.77 | 1.60 | 3.12 |
| Dasyprocta punctata | 3.4000 | 24 | 18 | 37 | 30 | 1 | 0.1088 | 0.0228 | 0.0363 | 0.0213 | 3.211 | 8.81 | 2.59 | 10.2 | 2.14 | 3.41 |
| Potos flavus | 3.9200 | 23 | 18 | 37 | 30 | 1 | 0.1657 | 0.0311 | 0.0211 | 0.0144 | 3.688 | 9.83 | 2.51 | 23.0 | 4.32 | 2.93 |
| Dasyprocta azarae | 4.1000 | 23 | 18 | 37 | 30 | 1 | 0.0935 | 0.0238 | 0.0304 | 0.0227 | 3.93 | 8.81 | 2.15 | 8.24 | 2.10 | 2.68 |
| Varecia rubra | 4.2000 | 23 | 18 | 36 | 30 | 1 | 0.0722 | 0.0357 | 0.0181 | 0.0224 | 4.052 | 8.23 | 1.96 | 6.45 | 3.19 | 1.62 |
| Alouatta sara | 4.4000 | 22 | 18 | 36 | 30 | 1 | 0.0812 | 0.0565 | 0.024 | 0.0099 | 4.228 | 8.77 | 1.99 | 16.4 | 11.41 | 4.85 |
| Monkey | 4.5000 | 22 | 17 | 36 | 29 | 1 | 0.11 | 0.042 | 0.023 | 0.021 | 4.304 | 9.49 | 2.11 | 10.5 | 4.00 | 2.19 |
| Martes pennanti | 4.7900 | 22 | 17 | 36 | 29 | 1 | 0.113 | 0.0412 | 0.0274 | 0.0211 | 4.588 | 9.88 | 2.07 | 10.7 | 3.91 | 2.60 |
| Trachypithecus vetulus | 5.0000 | 22 | 17 | 36 | 29 | 1 | 0.09 | 0.072 | 0.0192 | 0.0154 | 4.803 | 9.64 | 1.93 | 11.7 | 9.35 | 2.49 |
| Lutrogale perspicillata | 5.1000 | 21 | 17 | 36 | 29 | 1 | 0.152 | 0.0622 | 0.0485 | 0.0485 | 4.789 | 12.7 | 2.50 | 6.27 | 2.56 | 2.00 |
| Chlorocebus pygerythrus | 5.3000 | 21 | 17 | 35 | 29 | 1 | 0.089 | 0.0808 | 0.0426 | 0.0121 | 5.076 | 10.7 | 2.02 | 14.7 | 13.36 | 7.04 |
| Lutra lutra | 5.3300 | 21 | 17 | 35 | 29 | 1 | 0.255 | 0.0478 | 0.0514 | 0.0611 | 4.91 | 15.2 | 2.85 | 8.35 | 1.56 | 1.68 |
| Proteles cristata | 5.4000 | 21 | 17 | 35 | 29 | 1 | 0.182 | 0.0399 | 0.0906 | 0.0243 | 5.063 | 13.9 | 2.59 | 14.9 | 3.28 | 7.46 |
| Agouti paca | 5.4600 | 21 | 17 | 35 | 29 | 1 | 0.14 | 0.0321 | 0.0176 | 0.0222 | 5.248 | 10.5 | 1.92 | 12.6 | 2.89 | 1.59 |
| Macaca nigra | 5.6000 | 21 | 17 | 35 | 29 | 1 | 0.095 | 0.1052 | 0.0239 | 0.0186 | 5.357 | 10.9 | 1.96 | 10.2 | 11.31 | 2.57 |
| Puma yagouaroundi | 5.9000 | 21 | 17 | 35 | 29 | 1 | 0.116 | 0.043 | 0.0296 | 0.0391 | 5.673 | 11.4 | 1.93 | 5.93 | 2.20 | 1.51 |
| Hylobates concolor | 6.5500 | 20 | 17 | 35 | 29 | 1 | 0.293 | 0.1378 | 0.0582 | 0.0352 | 6.026 | 17.6 | 2.68 | 16.7 | 7.83 | 3.31 |
| Prionailurus viverrinus | 7.3000 | 19 | 16 | 34 | 28 | 1 | 0.16 | 0.0529 | 0.0335 | 0.0559 | 6.998 | 14.0 | 1.92 | 5.72 | 1.89 | 1.20 |



**Table A1.** *Cont.*

| Species | M | $q_{L,m}$ | $q_{Br,m}$ | $q_{H,m}$ | $q_{K,m}$ | $q_{R,m}$ | $m_L$ | $m_{Br}$ | $m_H$ | $m_K$ | $m_R$ | $q_{Het}$ | $q_{Het,m}$ | $m_L$ * | $m_{Br}$ * | $m_H$ * |
|---|---|---|---|---|---|---|---|---|---|---|---|---|---|---|---|---|
| Macropus agilis | 7.7000 | 19 | 16 | 34 | 28 | 1 | 0.203 | 0.0308 | 0.0602 | 0.0463 | 7.36 | 15.4 | 1.99 | 8.77 | 1.33 | 2.60 |
| Lontra canadensis | 7.9000 | 19 | 16 | 34 | 28 | 1 | 0.255 | 0.0425 | 0.0541 | 0.0747 | 7.474 | 17.1 | 2.17 | 6.83 | 1.14 | 1.45 |
| Dolichotis patagonum | 8.4300 | 19 | 16 | 34 | 28 | 1 | 0.158 | 0.0365 | 0.0651 | 0.036 | 8.134 | 15.0 | 1.78 | 8.78 | 2.03 | 3.62 |
| Symphalangus syndactylus | 8.5000 | 19 | 16 | 33 | 28 | 1 | 0.294 | 0.143 | 0.0515 | 0.0437 | 7.968 | 18.8 | 2.21 | 13.5 | 6.54 | 2.36 |
| Colobus guereza | 9.7500 | 18 | 16 | 33 | 28 | 1 | 0.171 | 0.0865 | 0.037 | 0.0233 | 9.432 | 15.6 | 1.61 | 14.7 | 7.42 | 3.18 |
| Felis chaus | 9.8000 | 18 | 16 | 33 | 28 | 1 | 0.153 | 0.0497 | 0.0483 | 0.0819 | 9.467 | 16.8 | 1.71 | 3.74 | 1.21 | 1.18 |
| Lynx canadensis | 10.000 | 18 | 16 | 33 | 28 | 1 | 0.158 | 0.0826 | 0.0388 | 0.0549 | 9.666 | 16.5 | 1.65 | 5.76 | 3.01 | 1.41 |
| Dog | 10.000 | 18 | 16 | 33 | 28 | 1 | 0.42 | 0.075 | 0.085 | 0.07 | 9.35 | 22.7 | 2.27 | 12.0 | 2.14 | 2.43 |
| Hystrix indica | 11.300 | 17 | 15 | 32 | 27 | 1 | 0.255 | 0.0407 | 0.0562 | 0.0524 | 10.85 | 18.8 | 1.67 | 9.73 | 1.55 | 2.15 |
| Theropithecus gelada | 11.400 | 17 | 15 | 32 | 27 | 1 | 0.236 | 0.1409 | 0.0772 | 0.038 | 10.91 | 20.3 | 1.78 | 12.4 | 7.42 | 4.06 |
| Pudu puda | 13.000 | 17 | 15 | 32 | 27 | 1 | 0.206 | 0.0616 | 0.0505 | 0.0199 | 12.56 | 18.4 | 1.43 | 20.7 | 6.19 | 5.08 |
| Gazella gazella | 15.000 | 16 | 15 | 31 | 27 | 1 | 0.327 | 0.0793 | 0.12 | 0.0406 | 14.43 | 24.6 | 1.64 | 16.1 | 3.91 | 5.91 |
| Castor fiber | 15.600 | 16 | 15 | 31 | 27 | 1 | 0.345 | 0.0489 | 0.044 | 0.0783 | 15.05 | 23.5 | 1.51 | 8.81 | 1.25 | 1.12 |
| Macaca arctoides | 15.900 | 16 | 15 | 31 | 27 | 1 | 0.241 | 0.118 | 0.061 | 0.05 | 15.4 | 22.9 | 1.44 | 9.64 | 4.72 | 2.44 |
| Lynx lynx | 17.5000 | 15 | 14 | 31 | 26 | 1 | 0.264 | 0.0943 | 0.093 | 0.0795 | 16.97 | 25.6 | 1.47 | 6.64 | 2.37 | 2.34 |
| Capreolus capreolus | 20.000 | 15 | 14 | 30 | 26 | 1 | 0.48 | 0.1 | 0.16 | 0.08 | 19.18 | 32.4 | 1.62 | 12.00 | 2.50 | 4.00 |
| Cuon alpinus | 20.000 | 15 | 14 | 30 | 26 | 1 | 0.346 | 0.116 | 0.158 | 0.0764 | 19.3 | 30.6 | 1.53 | 9.06 | 3.04 | 4.14 |
| Dog | 20.400 | 15 | 14 | 30 | 26 | 1 | 0.447 | 0.096 | 0.153 | 0.092 | 19.6 | 32.2 | 1.58 | 9.72 | 2.09 | 3.33 |
| Mandrillus sphinx | 23.000 | 14 | 14 | 30 | 26 | 1 | 0.331 | 0.168 | 0.076 | 0.0499 | 22.4 | 29.8 | 1.30 | 13.3 | 6.73 | 3.05 |
| Papio hamadryas | 23.300 | 14 | 14 | 30 | 26 | 1 | 0.392 | 0.174 | 0.103 | 0.0803 | 22.5 | 32.5 | 1.39 | 9.76 | 4.33 | 2.57 |
| Zalophus californianus | 34.000 | 13 | 13 | 28 | 25 | 1 | 1.274 | 0.31 | 0.168 | 0.2059 | 32 | 56.2 | 1.65 | 12.4 | 3.01 | 1.63 |
| Hydrochaeris hydrochaeris | 34.000 | 13 | 13 | 28 | 25 | 1 | 0.696 | 0.084 | 0.104 | 0.1035 | 33 | 42.2 | 1.24 | 13.5 | 1.62 | 2.01 |
| Canis lupus chanco | 38.000 | 12 | 13 | 28 | 25 | 1 | 0.971 | 0.14 | 0.303 | 0.2069 | 36.4 | 56.3 | 1.48 | 9.39 | 1.35 | 2.93 |
| Sheep | 52.000 | 11 | 12 | 27 | 24 | 1 | 0.96 | 0.106 | 0.28 | 0.16 | 50.5 | 61.7 | 1.19 | 12.0 | 1.33 | 3.50 |
| Reference women | 58.000 | 10 | 12 | 21 | 21 | 1 | 1.4 | 1.2 | 0.24 | 0.275 | 54.9 | 66.1 | 1.14 | 10.2 | 8.73 | 1.75 |
| Human | 60.000 | 10 | 12 | 21 | 21 | 1 | 1.7 | 1.3 | 0.32 | 0.25 | 56.4 | 72.2 | 1.20 | 13.6 | 10.4 | 2.56 |
| Reference man | 70.000 | 10 | 12 | 21 | 21 | 1 | 1.8 | 1.4 | 0.33 | 0.31 | 66.2 | 80.7 | 1.15 | 11.6 | 9.03 | 2.13 |
| Panthera tigris altaica | 75.000 | 10 | 12 | 26 | 23 | 1 | 1.1 | 0.342 | 0.305 | 0.4246 | 72.8 | 84.8 | 1.13 | 5.2 | 1.61 | 1.44 |
| Hog | 125.00 | 9 | 11 | 24 | 22 | 1 | 1.6 | 0.12 | 0.35 | 0.26 | 123 | 110 | 0.88 | 12.3 | 0.92 | 2.69 |
| Dairy cow | 488.00 | 6 | 9 | 21 | 20 | 1 | 6.46 | 0.4 | 1.88 | 1.16 | 478 | 354 | 0.73 | 11.1 | 0.69 | 3.24 |
| Horse | 600.00 | 6 | 9 | 20 | 20 | 0 | 6.7 | 0.67 | 4.25 | 1.66 | 587 | 458 | 0.76 | 8.1 | 0.81 | 5.12 |
| Steer | 700.00 | 6 | 9 | 20 | 19 | 0 | 5 | 0.5 | 2.3 | 1 | 691 | 434 | 0.62 | 10.0 | 1.00 | 4.60 |
| Elephant | 6650.0 | 3 | 6 | 15 | 16 | 0 | 6.3 | 5.7 | 2.2 | 1.2 | 6635 | 2327 | 0.35 | 10.5 | 9.50 | 3.67 |

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
