# Peer review of "Oxygen Deficient (OD) Combustion and Metabolism: Allometric Laws of Organs and Kleiber’s Law from OD Metabolism?"

_systems, doi:10.3390/systems9030054_

Round 1
Reviewer 1 Report
The present article proposed by Kalian Annamalai presents a "systems" approach involving the area of oxygen-deficient combustion (ODC) of a cloud of carbon particles and oxygen-deficient metabolism (ODM) and provides partial responses by treating each organ vital like a cellular cloud.
It is an interesting approach however the manuscript, to be accepted, must be proofread carefully to correct the multiple typographical and other imperfections (point where it is not necessary etc ...)
Figures should be reviewed like Figure 1, Figure 3, Figure 5 part of the information is not visible or half visible. The equations must be written with homogeneity (same font) which would make the journal more readable. Table 3 and Appendix A are also to be homogenized.
Reviewer 2 Report
The paper “Oxygen Deficient (OD) Combustion and Metabolism: Allometric Laws of Organs and Kleiber’s Law from OD Metabolism? by Kalyan Annamalai describes phenomena of some interest. A significant part of the paper looks like a review report. The allometric discussions compare phenomena in combustions and organisms. The fullness of topics is a source of confusion. Thus, reduction of text will lead to an improvement of the paper.
Reviewer 3 Report
The paper presents a pipeline that starts with a parallelism with automobile engines and biological species, for exploring then a “system” approach involving the field of oxygen deficient combustion (ODC) of a cloud of carbon particles and oxygen deficient metabolism (ODM).
The manuscript is very difficult to read, in the sense that it is difficult to follow the methodology, there is so much jargon and useless digressions. The document is not properly formatted, the figures are messy and with a very low quality and does not help in understanding the text. All these limit my understanding, and I feel myself unconforatble in truly capturing the main aim of the paper.
In general. the information within the manuscript are not original: it seems a review rather than an original article - with the only exception of paragraph 5.4.
Minor:
line 188, line 201, line 210, line 595: Error! Reference source not found.
Round 2
Reviewer 1 Report
This article can be accept in present form.
Reviewer 2 Report
The revised paper “Oxygen Deficient (OD) Combustion and Metabolism: Allometric Laws of Organs and Kleiber’s Law from OD Metabolism? by Kalyan Annamalai can be published - but it looks like a review.
Reviewer 3 Report
Thank you for addressing my suggestions. The paper looks more understandable now